# Integrated Optimal Design and Control of Fourth Generation District Heating Networks with Thermal Energy Storage

**Bram van der Heijde** [1,2,3], **Annelies Vandermeulen** [1,2,3], **Robbe Salenbien** [1,3] **and Lieve Helsen** [1,2,*]

1   EnergyVille, Thor Park 8310, 3600 Genk, Belgium
2   Department of Mechanical Engineering, KU Leuven, Celestijnenlaan 300, Box 2421, 3001 Leuven, Belgium
3   VITO NV, Boeretang 200, 5800 Mol, Belgium
*   Correspondence: lieve.helsen@kuleuven.be

**Abstract:** In the quest to increase the share of renewable and residual energy sources in our energy system, and to reduce its greenhouse gas emissions, district heating networks and seasonal thermal energy storage have the potential to play a key role. Different studies prove the techno-economic potential of these technologies but, due to the added complexity, it is challenging to design and control such systems. This paper describes an integrated optimal design and control algorithm, which is applied to the design of a district heating network with solar thermal collectors, seasonal thermal energy storage and excess heat injection. The focus is mostly on the choice of the size and location of these technologies and less on the network layout optimisation. The algorithm uses a two-layer program, namely with a design optimisation layer implemented as a genetic algorithm and an optimal control evaluation layer implemented using the Python optimal control problem toolbox called `modesto`. This optimisation strategy is applied to the fictional district energy system case of the city of Genk in Belgium. We show that this algorithm can find optimal designs with respect to multiple objective functions and that even in the cheaper, less renewable solutions, seasonal thermal energy storage systems are installed in large quantities.

**Keywords:** optimal design; optimal control; district heating; district energy systems; genetic algorithm; seasonal thermal energy storage; renewable energy

---

## 1. Introduction

Our energy system is one of the main contributors to the ever-increasing greenhouse gas (GHG) emissions, which calls for the identification of solutions within this sector. In particular, heating and cooling in residential and service buildings contribute to no less than 40% of the total final energy requirements in Europe [1]. Currently, 75% of the heating and cooling demand in buildings in the European Union (EU) (including industrial processes) is met by purely fossil resources [2], while of the remaining fraction 11% is provided by bio-mass, although these are usually polluting wood-stoves. Another 7% uses nuclear energy as a source through electricity and only the remaining 7% of heating and cooling come from 'truly' renewable sources, such as hydro, wind, solar and geothermal power. This makes the heating and cooling sector an ideal candidate to tackle the problem of both energy demand and GHG emissions in an efficient way. An often suggested solution is that of district heating and cooling systems to provide the heat and cold demand using centralised production with a large potential for reusing residual heat sources. Moreover, the most modern district heating and cooling systems (of the so-called *fourth generation*) allow for the inclusion of many low-temperature sources,

thermal energy storage (TES) and prosumers that inject heat or cold surpluses back into the network [3]. Dahash et al. [4] provided a comprehensive overview of large-scale thermal energy storage systems, concluding that although not necessarily the most cost-effective, tank and pit storage systems are often the most practical to install. They also found a clear gap in the research towards system integration of these seasonal storage systems in terms of modelling both accurately and in reasonable time.

In particular, Paardekooper et al. [5] calculated that switching to a large share of district heating in the European energy system—including only established technologies—enables a reduction by 86% of the $CO_2$ emissions compared to the levels of 1990 but also that district heating can cost-effectively provide at least half of the heating demand in 2050 in the 14 countries that are studied in the Heat Roadmap Europe projects. This reduction is the result of a switch to renewable and residual energy sources ($R^2ES$), enabled by the heat transport provided by district heating. Moreover, according to Lund et al. [6], (seasonal) energy storage will be needed in a highly interconnected energy system, namely to bridge the fluctuations in the availability of renewable energy sources. They calculated that thermal energy and fuel storage are by far cheaper technologies than electrical (battery) storage.

Although it is important to know the potential of district heating and cooling systems, particularly in combination with TES systems, these previous studies only describe fourth generation district energy networks in general terms. Hence, to realise these innovative energy networks, three challenges need to be overcome. Firstly, the design of systems with large shares of fluctuating renewable energy sources will be much more complicated than that of present thermal networks. Secondly, identifying and implementing the right control strategy for a given network will be harder as well, due to fluctuating energy sources and large shares of energy storing components in the network, including energy flexibility. The third challenge follows from the first and second, namely the fact that the choice of control strategy will influence the optimal design and vice versa. This calls for an integrated strategy in which control and design are concurrently optimised.

### 1.1. Previous Studies on District Energy System Design

Söderman and Pettersson [7,8] made a topology optimisation algorithm for district energy systems (DES). The algorithm was based on a mixed integer linear program (MILP) for a district including thermal and an electric grid. Thermal energy storage was included in the optimisation, too. They limited the problem to eight representative time instances, namely typical daily and nightly operation conditions in the four seasons. Weber [9] integrated the optimisation of both design and control of poly-generation systems in DES with different energy carriers, but without considering TES. Again, the temporal detail remained limited. Weber used a bi-level solution strategy, where a master optimisation (evolutionary algorithm) chose the type, size and location of technologies to be installed in the network. The slave optimisation (mixed integer non-linear program) decides the layout of the network and the operational strategy.

Fazlollahi et al. [10] presented a multi-objective, non-linear optimisation strategy for DES including district heating and poly-generation, but without considering large-scale TES systems. They used a problem subdivision similar to that of Weber, where a master evolutionary algorithm varies the design parameters, and the proposed designs are evaluated by an MILP, which optimises the energy flows during 8 typical periods. Fazlollahi implemented an additional layer for the thermo-economic optimisation, and a post-processing step to assess the emissions of the proposed designs. The optimal results were summarised in Pareto-fronts according to system efficiency, total annual system cost and $CO_2$ emissions. In this study, the district heating supply and return temperatures were varied as a function of the ambient temperature.

Other studies combine the entire optimisation in a single mathematical problem, often an MILP. Dorfner and Hamacher [11] used this strategy to find the optimal lay-out and pipe size of district heating networks in Germany. This study omitted the operational aspect, instead only considering peak loads. Morvaj et al. [12] presented a single optimisation problem integrating design, operation and network layout for an urban energy system with 12 buildings. They considered one representative

day for each month, averaging the electric and heat load profiles for a whole year. Falke et al. [13] presented a similar multi-objective optimisation problem as Fazlollahi and Weber, but they considered a rule-based control flow for the operational layer, as opposed to an optimal control strategy.

In a wider energy system context, Patteeuw and Helsen [14] presented an integrated control and design optimisation algorithm for the design of the space heating and domestic hot water production system for residential buildings in the Belgian energy system, assuming a number of scenarios for the composition of the future electricity system. They used a single-layer MILP optimisation algorithm with representative weeks to reduce the temporal complexity of the optimisation problem. However, they found that this approach is very slow. They suggest that a scenario-based optimisation is more efficient than a full optimisation problem in which the scenario parameters are included as decision variables. This suggestion clearly points in the direction of a two-layered optimisation approach.

Lund and Mohammadi [15] presented a methodology to optimise the choice of insulation standard for pipes in thermal networks. Their method is split in two calculation tools: one to calculate different scenarios of heat loss behaviour in the thermal network, and the other where the energy flows in the larger system are optimised using EnergyPlan. An evolutionary design algorithm was coupled to EnergyPlan as the evaluation problem by Prina et al. [16]. Their focus was on the operation of regional energy systems to find both techno-economically feasible, as well as sustainable energy system designs. In a later step [17], they accounted for the long-term investment planning problem, considering the evolution of the price for different technologies and the remaining value of previously installed systems as they are replaced by more modern technologies.

Bornatico et al. [18] used a particle swarm optimisation (PSO) algorithm to optimise the thermal system of a Swiss single residential building (hence no DES or thermal network was considered). They aimed to optimise the size of a solar heating system, including a solar collector, storage tank and auxiliary power unit. In this study, the system was simulated in *Polysun*, coupled to MATLAB for the PSO implementation. Whether Polysun implements a heuristic or optimal control was not specified. The results of the PSO were compared to a genetic algorithm and the results were found to be similar. Ghaem Sigarchian et al. [19] optimised a hybrid microgrid including solar photovoltaic panels and concentrated solar power collectors, an organic Rankine cycle to convert heat to electricity, electric and thermal energy storage and a gas-fired backup generator. Both design variables (in the PSO) and a variable operation (in HOMER) were considered. The objective function was the energy tariff to be paid by the consumers in the network, which had to be minimised. The fitness evaluation function seems to be an optimal control problem implemented in HOMER, although this is not clearly explained.

In conclusion of the previous work, a clear pattern is that the optimisation algorithm is subdivided in two layers, where one layer is aimed at evaluating the operational aspect of a particular design—the lower layer or *slave algorithm*—and the other focusses on the exploration of the design parameter space—the upper layer or *master algorithm*. While there are subtle variations where for instance the slave algorithm also optimises part of the design variables, this general structure holds for most of the above discussed references. Still, a smaller number of studies use a completely integrated control and design algorithm, with a single layer that optimises both operational variables and design parameters. Clearly, this approach represents only a minority in the discussed studies and is only suited for design problems with a limited size and (temporal) complexity.

### 1.2. Novelty and Contribution

The aim of this paper is to develop an integrated design and control optimisation algorithm for future district heating systems with large shares of R$^2$ES and seasonal thermal energy storage. This algorithm is illustrated in a fictitious district heating system for an existing city in Belgium and the design results from the optimisation are studied in detail. Note that the focus is on the methodological contribution, rather than on the absolute numbers resulting from the case study.

Compared to pre-existing studies, this paper uses a two-layer approach, focussing on the integration of a higher-resolution full-year optimal control problem (OCP) as the lower-level

optimisation layer, with particular attention paid to the high operation variability of future energy systems with distributed energy resources. In order to do so, a Python toolbox called `modesto` (see Vandermeulen et al. [20]) is used to set up these OCPs. We use an optimal selection of representative days compatible with seasonal thermal energy storage systems to reduce the calculation time. To our knowledge, this is also the first study in which a concurrent design of TES volumes, pipe diameters and heat generation systems is considered, together with a more detailed model of the district heating system.

## 2. Methodology

The optimisation framework is conceived as a two-layer integrated optimal design and control algorithm. In Section 2.1, the heat demand for space heating and domestic hot water is calculated deterministically and used as a fixed boundary condition for the algorithm. The slave optimisation is a linear optimisation which determines the optimal energy flows in the network for a given design, including the TES charging behaviour, implemented in `modesto`. This layer of the algorithm is explained in Section 2.2. The master optimisation is a genetic algorithm which looks for the optimal combination of design parameters, based on a number of objective functions. This layer is described in Section 2.3. Apart from the implementation, this section also summarises the available design choices for the chosen case study, as well as the considered scenarios.

### 2.1. Case Study

The optimisation algorithm is illustrated by means of a fictitious DES for the city of Genk in Belgium, called *GenkNet*. Spread over 9 neighbourhoods, 7775 building models were constructed based on geometric data for single family residential buildings. Although the network configuration and the choice of the connected neighbourhoods are hypothetical, the data with which the building models were constructed are real.

The building models are equivalent resistance-capacitance models based on the `TEASER` `FourElement` structure (see Remmen et al. [21]). Assumptions on the building materials and wall thicknesses were based on the building age, which was assumed fixed for all buildings belonging to one neighbourhood. The workflow to derive the building model parameters was developed by De Jaeger et al. [22]. The heat demand resulting from space heating and domestic hot water (DHW) production was simulated using a typical meteorological year for Belgium and stochastic occupant profiles as boundary conditions. The occupant profiles contain the space heating temperature set point and the DHW draw-off for every individual building and were derived using the `StROBe` toolbox (see Baetens and Saelens [23]). An ideal building heating system (neglecting the effects of the heating system temperatures on the heat injection in the building) was assumed. All buildings were simulated during a full year with a 900 s time step using a minimum energy objective (assuming a fixed cost for heat), after which the heat demand of all buildings belonging to one neighbourhood was summed and modelled as a single demand node in the network. The heat distribution network on the neighbourhood level is omitted, which means an underestimation of the total heat losses in the network. Instead, the neighbourhood is represented as a single node, connected to the backbone network through a single service pipe.

The resulting heat demand for the 7775 buildings amounts to 430.5 GWh per year, with an average energy use intensity of 210.8 kWh/m$^2$ per year. Cooling and electricity demand were not considered in this study.

The neighbourhoods were located alongside a central thermal network backbone, as indicated in Figure 1.

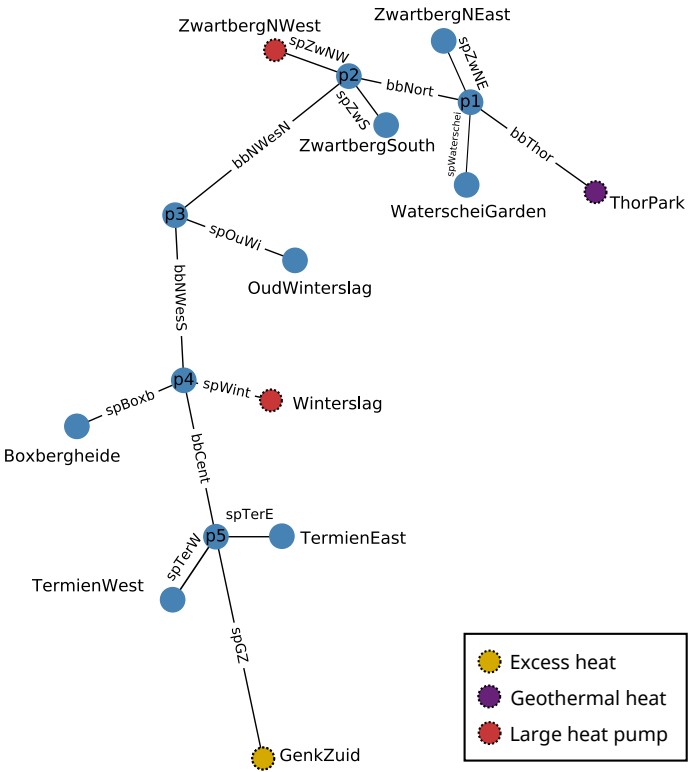

**Figure 1.** Network layout for the fictitious case study in Genk, Belgium. The heat demand of 9 neighbourhoods alongside a central backbone connection was aggregated per neighbourhood. Two additional nodes without heat demand, but with the option to install heat sources and/or thermal energy storage (TES) systems are situated at both ends of the backbone.

## 2.2. Operational Optimisation

We have chosen for a full-year OCP with a 2 h time-step for the evaluation of the DES performance with respect to a number of objective functions (see Section 2.3). The OCP optimises heat and mass flows in the network, the operation of the heat production systems and the charging behaviour of the TES systems in order to satisfy the heat demand of the neighbourhoods. This optimisation has a single objective function to minimise the operational cost. The choice for a minimal operation cost objective is based on the habit of operating real systems to maximise their profit. Except for a limited number of experimental systems, systems are seldom operated to minimise energy use or $CO_2$ emissions, unless this is linked to additional economic incentives. The choice for an optimal control strategy as opposed to a simulation based evaluation or a rule-based control strategy is justified by the quantification of the maximum potential of every design. This potential is always reached when we assume an optimal control strategy exists and can be implemented, whereas heuristic control strategies might penalise designs that are harder to control. As such, we arrive at a fair comparison of designs.

This OCP is implemented using the Python toolbox `modesto` (see Vandermeulen et al. [20]). This toolbox is built on top of Pyomo [24] and implements a library of linear optimisation models for common DES components and communicates with an optimisation solver to find the optimal operation strategy. More details about the used component models can be found in Appendix A. All models were either derived from literature or verified by the authors. The optimisation variables considered in the operational layer are the magnitude of heat and mass flows in all components, the thermal output of heating systems, possible curtailment of heat from the solar thermal collectors and the state of charge of the TES systems.

The solution time of the OCP is reduced by using representative days. Van der Heijde et al. [25] have developed a method to optimally select representative days and to restore the chronology such

that the original data is approximated as closely as possible. This method is applied here. Based on a number of input time series, specified by the user, an optimal set of representative days is chosen, after which the algorithm determines for each day of the year which representative day it will be represented by. In this work, the chosen input time series were the aggregated heat demand for all neighbourhoods, the solar radiation on a unit surface area, the ambient temperature and the hourly electricity price. This method furthermore makes sure that seasonal effects in the TES systems are modelled accurately. We have limited the representative day selection to 12 days as this was shown to be sufficient to represent the full-year OCP with acceptable accuracy, see the conclusion made by van der Heijde et al. [25].

### 2.3. Design Optimisation

The design parameter values are varied in the upper layer. While we attempted to keep the slave optimisation linear, both to guarantee a global optimum and to limit the calculation time, non-linear effects do appear in the master optimisation problem. These non-linearities include: investment costs, which can vary with the size of the installed system; discrete decision variables, such as pipe diameters; and the calculation of the state-dependent heat loss from thermal storage tanks. This last phenomenon is caused by the use of a TES model in which the heat loss depends on the actual state of charge. On the other hand, this loss fraction also depends on the size of the TES system, which would render the optimisation problem bi-linear (see the derivation by Vandewalle and D'haeseleer [26]). To avoid this extra non-linearity, the design variable (namely the size of the TES systems) is treated as a constant parameter in the OCP and it is varied in the master optimisation.

We implemented the design optimisation algorithm as a genetic algorithm in Python using the DEAP (Distributed Evolutionary Algorithms in Python) toolbox [27]. The algorithm uses the NSGA-II (Non-dominated Sorting Genetic Algorithm II) selection operator [28]. Crossovers are handled by a simulated binary crossover operator, and for mutation, a polynomial bounded operator is used. Moreover, the genetic algorithm features a small probability of entirely reinitialising some parameters, which is a variation on the mutation operator. The Pareto-optimal solutions of all generations are stored in a *Hall of Fame*. Every new generation is initialised based on all non-dominated individuals, taken over all previous generations. As such, previous optimal solutions cannot be lost in the course of the evolution. A single optimisation run features 100 generations with 60 individuals. Each newly generated individual has a 95% mutation probability and a 70% crossover probability.

Every candidate design is evaluated as an instance of `modesto` with a minimal cost objective (see Section 2.2). Infeasible optimisation problems result in a high penalty objective value, such that these designs are not selected in the next generation. With `modesto`, the optimal control trajectory for all energy and mass flows in the network during one year is computed, such that the operational cost is minimal. The workflow of this genetic algorithm is illustrated in Figure 2.

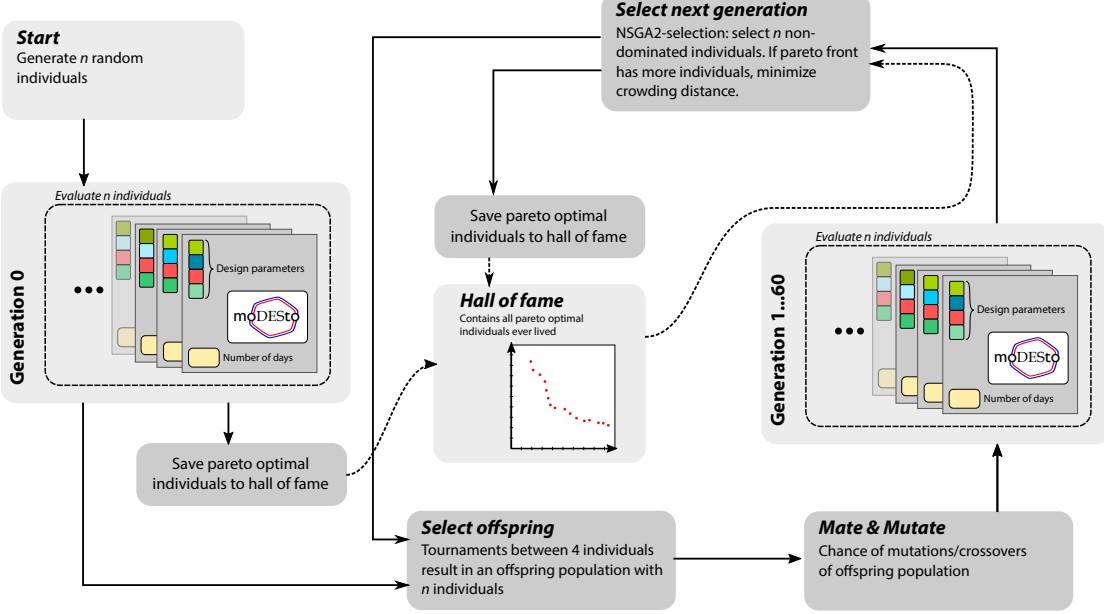

**Figure 2.** Flow chart illustrating the different steps in the genetic algorithm.

### 2.3.1. Objective Functions

The design was evaluated for optimality based on three objectives, namely the annual primary energy imported from outside the district energy system—used for the heat pumps, geothermal heating plant, possibly auxiliary heating needed for the production of DHW and network pumping power—, the total annualised costs and the $CO_2$ emissions. The total annualised cost $c_a$ consists of the annual operation cost $c_{op,a}$ as calculated by `modesto`, increased by the annualised investment $I_a$ and fixed annual maintenance cost $c_{maint,a}$ :

$$c_a = I_a + c_{op,a} + c_{maint,a}. \tag{1}$$

The annualised investment cost $I_a$ is calculated using the capital recovery factor:

$$I_a = I_{tot} \frac{(1+i)^\tau i}{(1+i)^\tau - 1}, \tag{2}$$

where $I_{tot}$ stands for the total investment, and $I_a$ for the annualised investment. $\tau$ denotes the economic lifetime of the technology and $i$ is the interest rate, taken as 3%, assuming a public investment on a long term. This is in line with multiple studies, such as that of Möller and Werner [29], Nussbaumer and Thalmann [30] and Steinbach and Staniaszek [31]. The annualisation calculation using the capital recovery factor assumes that every component is replaced by an identical system at the end of its lifetime, and at the same investment. As such, variations in technology prices over time are neglected.

For maintenance ($c_{maint,a}$), the annual contribution is estimated to be a fixed fraction of the initial investment. This fraction, as well as the typical economic life time of various technologies, were derived from the EnergyPlan Cost Database [32]. All economic input data for the different technologies considered in this work is summarised in Appendix B.

In order to get a better grasp of the orders of magnitude of the objective functions—that is, annualised total costs, primary energy import and $CO_2$ emissions—we scale them with respect to the total annual heat demand of all neighbourhoods for space heating and DHW. This total heat demand amounts to 430.5 GWh per year. The resulting scaled variables are called the levelised cost of heat

(LCOH, expressed in EUR/kWh), the primary energy import share (PEIS, in %) and the $CO_2$ intensity (in kg $CO_2$/kWh).

### 2.3.2. Design Choices

The *GenkNet* case has a total of 9 neighbourhoods, 1 industrial node and 1 node with additional heat generation systems but no demand. The design exercise is left very open; the optimiser has to choose how many renewable resources for heat generation are installed at every node, where the TES systems are installed and how large they should be, and how much backup power is needed. The available design choices for the TES systems are listed in Table 1, the solar thermal collector (STC) arrays in Table 2, and the backup heat pumps and geothermal heating plant in Table 3. The maximum volume corresponds to the largest pit and tank TES systems currently found in literature. The maximum STC area corresponds to the available south-oriented roof area of the buildings in the neighbourhoods, however without accounting for previously installed systems, such as PV panels. At node *ThorPark*, a larger area is assumed to be available for the installation of an STC array.

**Table 1.** Available design choices for TES systems at the different nodes in GenkNet. All numbers are expressed in m$^3$.

| Node | Component | Min | Max ($\times 10^3$) |
|---|---|---|---|
| Boxbergheide | PTES | 0 | 200 |
| OudWinterslag | PTES | 0 | 200 |
| TermienEast | PTES | 0 | 200 |
| ThorPark | TTES | 0 | 12.5 |
| WaterscheiGarden | PTES | 0 | 200 |
| Winterslag | PTES | 0 | 200 |
| ZwartbergNEast | PTES | 0 | 200 |
| ZwartbergNWest | PTES | 0 | 200 |
| ZwartbergSouth | PTES | 0 | 200 |

**Table 2.** Available design choices for the installed STC array area per node. All nodes except *ThorPark* consider the total available South-oriented rooftop area of the considered buildings in that neighbourhood. All numbers are expressed in m$^2$.

| Node | Min | Max ($\times 10^3$) |
|---|---|---|
| Boxbergheide | 0 | 78.5 |
| OudWinterslag | 0 | 15.5 |
| TermienEast | 0 | 12.9 |
| TermienWest | 0 | 16.3 |
| ThorPark | 0 | 100.0 |
| WaterscheiGarden | 0 | 49.9 |
| Winterslag | 0 | 37.8 |
| ZwartbergNEast | 0 | 12.7 |
| ZwartbergNWest | 0 | 17.0 |
| ZwartbergSouth | 0 | 33.3 |

**Table 3.** Design choices for the nominal power of the central heat generation systems. The power is expressed in MW. The abbreviations "geo" and "hp" stand for geothermal heating plant and air source heat pump, respectively.

| Node | Component | Min | Max |
|---|---|---|---|
| ThorPark | geo | 0 | 40.0 |
| Winterslag | hp | 0 | 80.0 |
| ZwartbergNWest | hp | 0 | 80.0 |

In addition, the pipe diameters are also design decision variables. For the available diameters, the reader is referred to Tables A2 and A3. The smaller diameter pipes are implemented as twin pipes (up to DN 200), the larger pipe diameters as compound pipes. The investment costs for these pipes are discussed in Appendix B.3. The available diameters were derived from IsoPlus [33]. It is also an option to install no pipe at all at a specific network edge; this choice is represented by a 0 m diameter pipe.

All scenarios share the same network layout, as shown in Figure 1. Whereas the choice for the size of the heat pumps and the geothermal heating plant is handled by the optimisation algorithm, the availability of excess heat is fixed at 10 MW, constantly available throughout the year. Whether this resource is utilised or not is a question of the decision of the district heating connection between the node *GenkZuid* and the rest of the network, that is, is it worth the investment to make a connection to the industrial area from the city or not.

### 2.3.3. Scenarios

While most of the boundary conditions are fixed for the design algorithm, two of them are varied discretely and deterministically to establish their influence on the results, leading to a number of scenarios. The first boundary condition is the nominal temperature level in the network. Four options are available:

- 45–25 °C,
- **55–35 °C** *(base scenario)*,
- 65–45 °C and
- 75–35 °C.

Hence, most scenarios use a 20 K nominal $\Delta T$ with rather low supply temperatures, whereas the last scenario uses medium-high temperatures with a 40 K nominal $\Delta T$.

The second scenario parameter is the cost of heat from the industrial excess heat source in the most southern node of the network. The heat prices considered are:

- 5 EUR/MWh,
- 10 EUR/MWh,
- **15 EUR/MWh** *(base scenario)* and
- 20 EUR/MWh.

These excess heat costs are substantially higher than the ones discussed by Doračić et al. [34], but they are chosen to be in line with the expected cost for an industrial company that needs to invest in a connection of its processes to a district heating system.

The different combinations of the two scenario parameters lead to a total of 16 optimisation runs to be performed. However, the focus will be on how the scenarios deviate from the reference scenario—55/35 °C with 15 EUR/MWh excess heat cost—leading to a total of 7 scenarios to be studied in detail.

## 3. Results

The emphasis of this paper is on the methodological contribution, namely the integrated design and control optimisation algorithm. The results presented in this section should mostly be interpreted as a proof of concept, rather than in absolute numbers.

### 3.1. Reference Case Results

As mentioned before, the case with a 55/35 °C temperature regime and an excess heat cost of 15 EUR/MWh is chosen as the reference case. This section shows a selection of visualisations of the optimal design results in order to make more sense out of the large amounts of output data. Firstly, we focus on the higher level, using only design parameters and yearly aggregated outcomes, but in a later stage we will also zoom in on the results on smaller time scales.

Genetic Algorithm Outcome

The solutions resulting from the genetic design algorithm, ranked by the objectives of LCOH and PEIS are shown in Figure 3. On the one hand, this figure shows the spread of all considered designs that turned out to be feasible in terms of their LCOH and PEIS values. The blue dots show the Pareto-optimal solutions, that is, the solutions that dominate the solution space. The three black markers respectively show the solutions with the lowest PEIS, the one with an approximately average PEIS measured over all non-dominated solutions and the one with the maximal PEIS. These three specific solutions will be treated in more detail in the next subsection. Note that the minimal and maximal PEIS solutions also represent the respective maximal and minimal LCOH solutions.

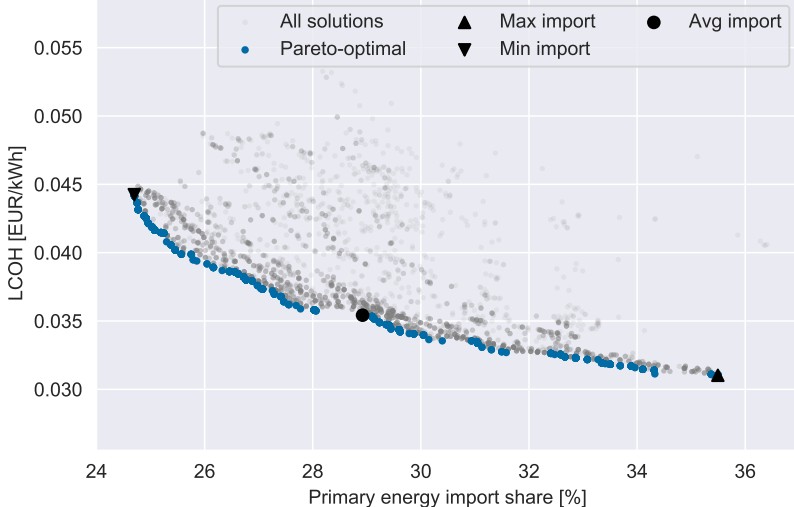

**Figure 3.** Scatter plot of all genetic design algorithm solutions for the reference design case.

Zooming in on the Pareto-optimal solutions only, we can plot the contributions of operational costs, investment and maintenance costs to the LCOH. This cost breakdown is presented in Figure 4, where we see that the contribution of the annualised investment takes up the largest share of the total annual costs of the system. As expected, as a larger amount of energy is imported from outside the network, the operational costs (representing in this case the cost of electricity use) increase linearly.

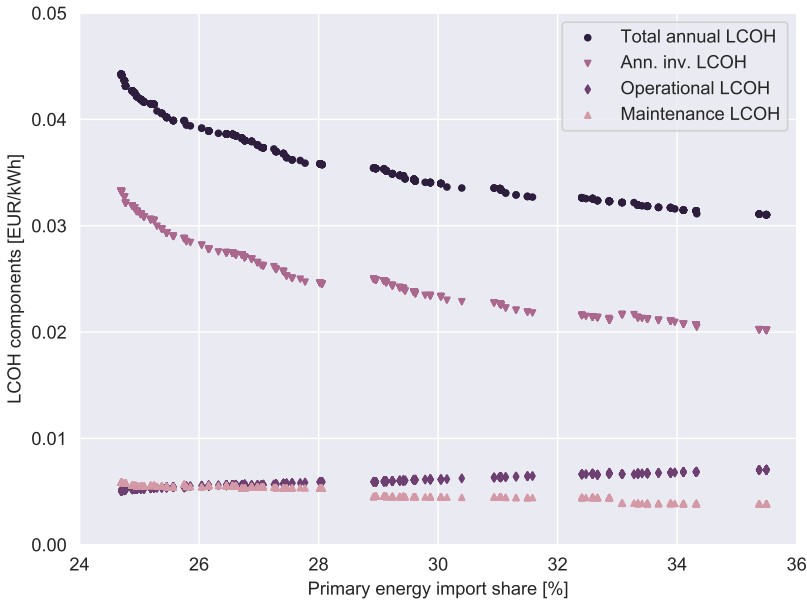

**Figure 4.** Different contributions to the total annual LCOH for the reference case, namely that of annual maintenance, operational cost and annualised investment cost. The annualised investment has the largest contribution.

The contribution of the different technologies is shown in Figure 5, showing indeed that the higher investment for the lower PEIS solutions is largely caused by a larger installed amount of TES and STC systems. The bar chart furthermore shows that the investment in auxiliary heating plants (i.e., the heat pumps and the geothermal plant) combined remains more or less constant. This does not mean that the installed auxiliary power remains constant as well, given the different prices per unit of thermal power for the two technologies. Finally, Figure 5 shows that the investment in the transport pipes remains more or less constant, too. On closer inspection (not shown here), we find that the lower PEIS solutions have marginally larger investments in the network, corresponding to wider pipes on average.

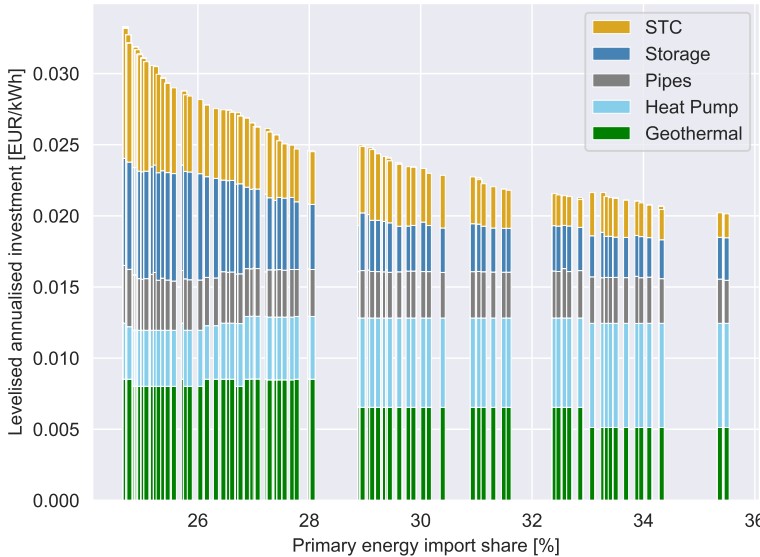

**Figure 5.** Bar chart showing the contribution of different technologies to the levelised annualised investment costs. The cost of the auxiliary domestic hot water (DHW) boilers has been omitted since it is the same for all solutions shown.

Figure 6 shows the distribution of the chosen sizes for the STC and TES systems in the network, split by network node and *energy range*. The energy range is a categorisation of the Pareto-optimal solutions based on their PEIS. The range of PEIS values for all Pareto-optimal solutions is split in three equal parts, denoting high, mid and low energy import. The plot shows three violin plots per neighbourhood and parameter, indicating the approximate distribution, as if the design parameters were distributed following some probability function. The plot gets wider where there is a denser distribution of observations for that parameter. The actual parameter values are indicated by the black dots inside the violin plots, which shows that the distribution is in fact very sparse, with a few dense spots here and there.

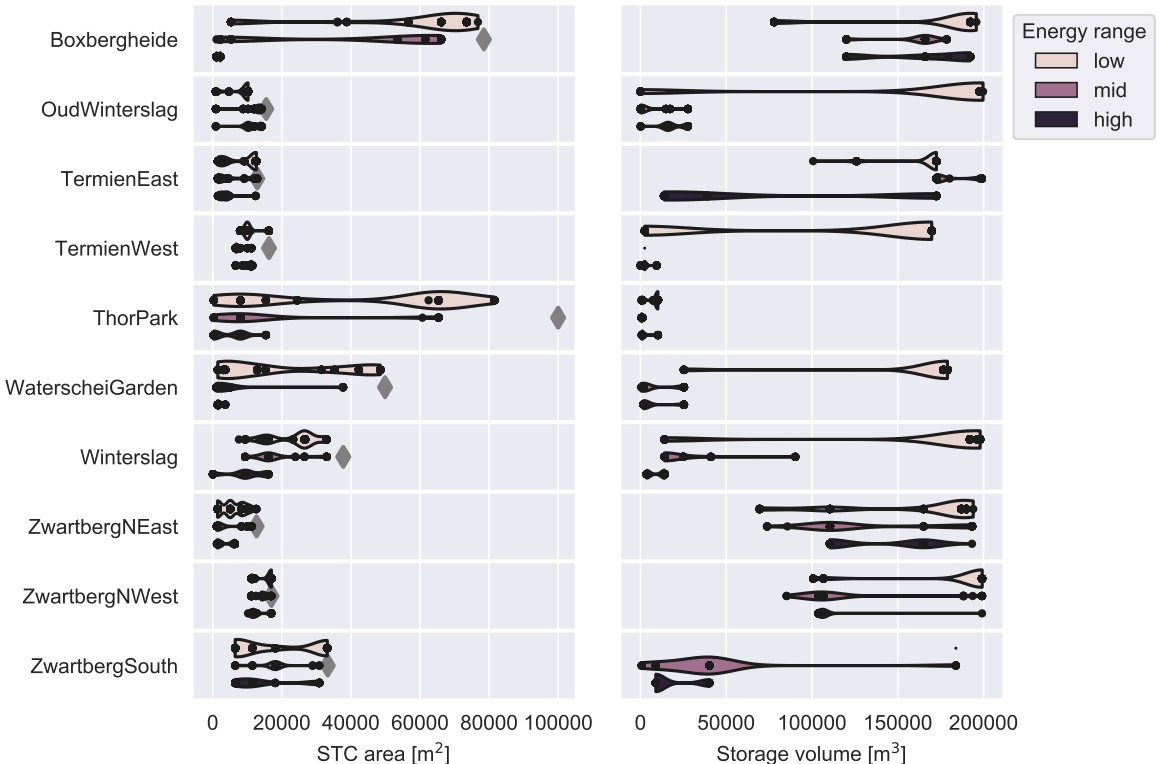

**Figure 6.** Distribution of solar and storage systems per network node. The Pareto-optimal solutions are split into high, mid and low energy import solutions. The left graph shows the maximum STC area per neighbourhood as the grey diamond.

The solutions with lower PEIS are expected to have a larger share of solar energy, which is only possible if enough STC area is installed, as well as TES volume. For example, *Boxbergheide*, *ThorPark*, *WaterscheiGarden* and *ZwartbergNEast* show this evolution of decreasing STC area with increasing imported energy. For the other neighbourhoods, the spread stays largely the same, although in most cases the average shifts to lower values. Only for *OudWinterslag*, no such shift can be distinguished. Another interesting observation is that more often than not, almost the maximum available area is exploited for the installation of STC systems.

Looking at the TES volumes, such a trend is not so easy to find. In the case of *OudWinterslag*, *TermienWest*, *WaterscheiGarden*, *Winterslag* and *ZwartbergSouth*, we can see that at least the low energy range has the largest storage volumes. However, if we compare this graph with the map (Figure 1), one interesting trend is that larger storage tanks tend to be installed as close as possible to locations where auxiliary heating plants (i.e., heat pumps and geothermal) are available. This is more obvious for the neighbourhoods of *ZwartbergNWest* and *Winterslag-Boxbergheide*. *ZwartbergNEast* seems to act as the storage hub for *ThorPark*, given the smaller distance between these nodes than the distance between

*Waterschei Garden* and *ThorPark*. The reason for the predominantly large volume at *ZwartbergNEast* can be explained by the limited storage potential at the *ThorPark* node.

When the distribution of the nominal power of the production systems is plotted per neighbourhood, we find that the variety in the solutions is rather low. Figure 7 shows that the size of the heat pump in *ZwartbergNWest* is positively correlated with the energy import range and the opposite is true for the geothermal heating plant. The heat pump at *Winterslag* has an almost constant nominal power.

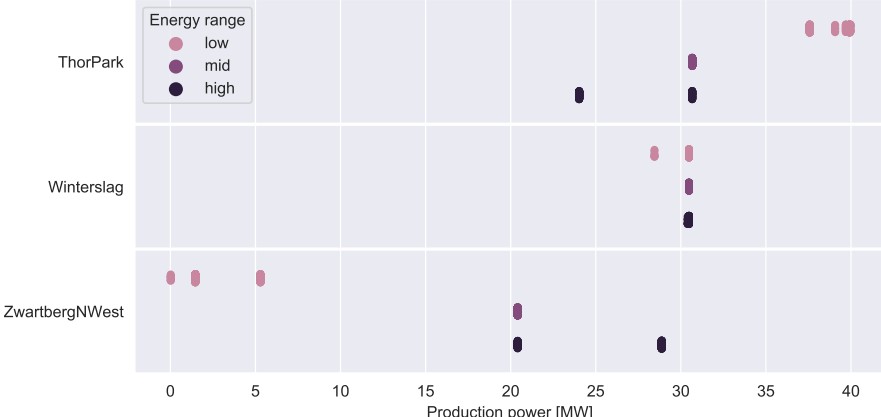

**Figure 7.** Spread of the nominal power of the chosen production systems, again subdivided according to the range of the energy import from outside the network. The heat pump at *Winterslag* is nearly always the same size, at *ZwartbergNWest* the size increases with the range of energy import and the geothermal heating plant decreases with increasing energy range.

A final plot of interest is the spread of the sum of all storage volumes compared to the total STC area in the network. This graph is shown in Figure 8, and a sigmoid distribution appears clearly. For lower STC areas, the storage volume appears to increase linearly with the STC area, until around $150 \times 10^3 \, \text{m}^2$, where the storage volume starts increasing very rapidly. As soon as the maximum storage volume is reached, the STC area can still increase, but mostly at a higher cost without much reduction in terms of import share.

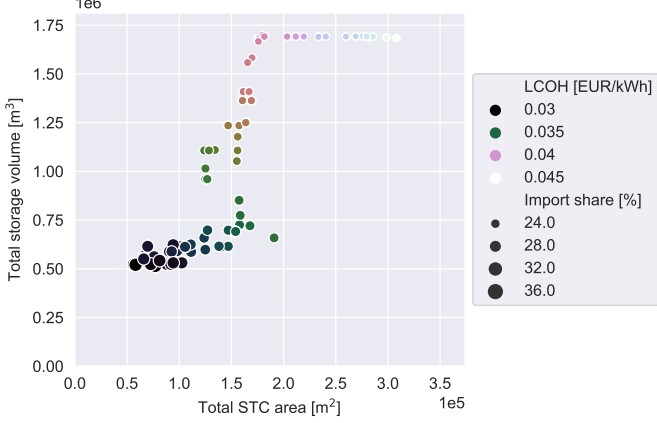

**Figure 8.** Distribution of total STC area compared to the total storage volume. The scatter points are coloured according to the LCOH, with darker values representing a lower LCOH than the lighter values. The larger the dot, the larger the import share (and thus the larger the grid dependence of the network). Note that the axes are limited by the minimum and maximum allowed design sizes.

## 3.2. Detailed Study of Highlighted Reference Solutions

In Figure 3, three particular solutions are indicated with a black marker. Two solutions respectively at the upper and lower end of the Pareto front were chosen, and one solution that is closest to the average PEIS of all Pareto-optimal solutions. For these designs, we investigate the heat sources in the network and the energy storage levels in more detail.

### 3.2.1. Maximum LCOH, Minimum PEIS

The first highlighted solution is the one with the lowest primary energy import from outside the network, but the highest LCOH. In the graph of the heat flow rates (the upper subplot in Figure 9), the black line indicates the part of the heat demand of the neighbourhoods which is supplied by the district heating network. We see that the excess heat production is used as a base load throughout most of the year, except for the Summer period, where most of the heat demand is met by a combination of solar thermal power and discharging of the TES systems.

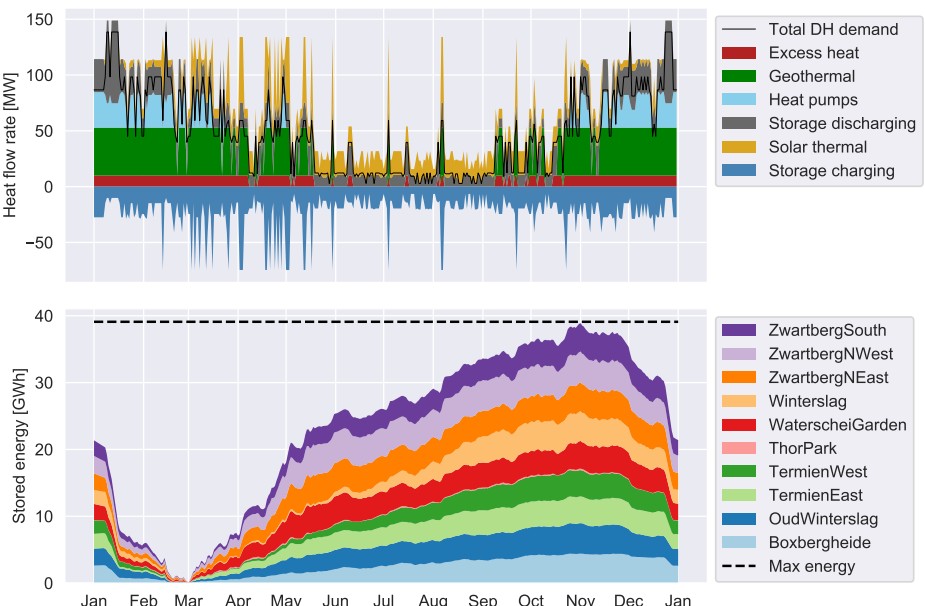

**Figure 9.** Full year time series for minimum PEIS design.

The geothermal plant is runner-up after the excess heat injection. Its operation is fixed by the installed nominal heating power and the fixed operation schedule throughout the year. Additional gaps are filled mostly by the heat pumps and by discharging the storage systems. The STC panels inject heat whenever they can, and surpluses are stored for later use. This is visible from the part below the zero power level, which indicates the charging behaviour of all tanks combined.

When the net power graph is studied in a shorter time range (see Figure 9), we can clearly see that power exceeding the neighbourhood heat demand is stored in the TES systems.

Around 40 GWh of TES is installed, and a clear seasonal charging pattern is obvious from the lower subplot in Figure 9. At the start of March, the storage tanks are completely empty, and during Spring and Summer they are gradually charged with energy until the maximum storage level is reached in November. Then, the storage is quickly depleted until the cycle repeats. A result of the genetic algorithms optimisation is that the full storage volume is used. Solutions where the charged energy profile does not use the full available range are dominated by more efficient designs.

An interesting result is that most storage tanks are used with a more or less similar charging pattern, witnessed by the evenly spread contributions of different systems. Only the smaller *ThorPark* TTES system is overshadowed by the other systems.

Figure 10 zooms in on a winter, spring and summer week to show a more detailed image of the heating power flows in the system. In the winter week, we see that the geothermal plant is working continuously, largely assisted by the heat pumps at their full power.

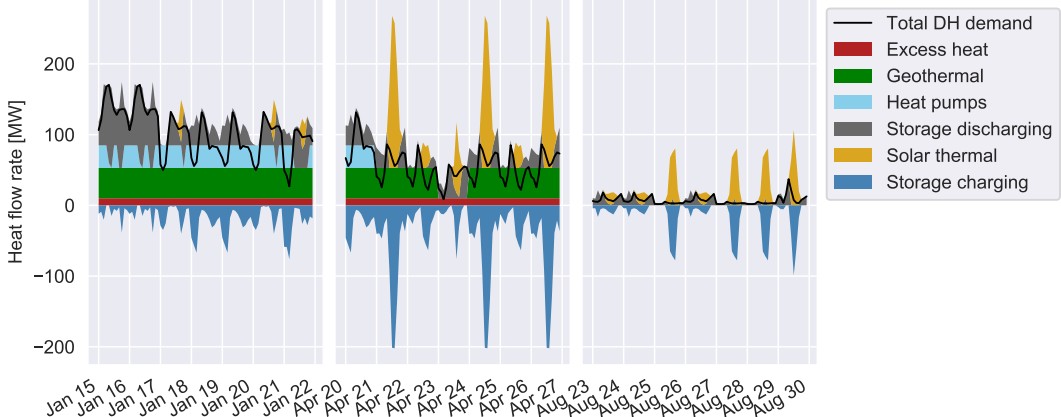

**Figure 10.** Detailed heating power plot of the minimum PEIS design for a winter, spring and summer week.

The remaining heat demand is met by discharging the TES systems. During the spring week, we see the intermittent behaviour of the geothermal plant (which is fixed by the off- and on-periods as determined by the user), but also the high power coming from the STC systems. Note that the TES tanks are mostly charging and only limited TES discharge is needed to fill some gaps in the heat demand. The heat pump is only on for higher demand levels. Note that the maximum amount of excess heat is being injected during both weeks (winter and spring). Finally, the summer week shows a very low heat demand, and while all "auxiliary" conversion systems are off, the demand is met by discharging the storage and power from the STC systems. Note how the surplus of solar heat is charged to the TES systems.

### 3.2.2. Intermediate LCOH and PEIS

The intermediate solution (see Figure 11) shows a rather different picture: whereas the previous solution was characterised by heat flow rate peaks over 200 MW, here the peak powers are more limited. The positive peaks do not exceed the maximum heat demand, which seems to suggest that the network has been designed to accommodate the maximum heat demand and not more. A slightly smaller geothermal system seems to be suggested by the power graph, but this is compensated by larger heat pumps. The lower positive peaks show that a smaller STC area has been installed.

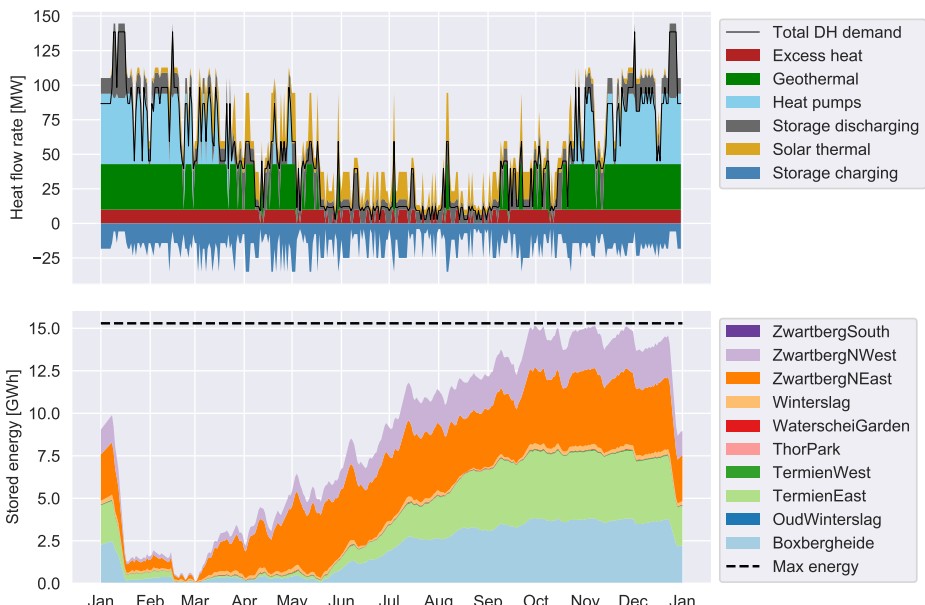

**Figure 11.** Full year time series for intermediate PEIS design.

The size of the TES systems is also considerably smaller, peaking at energy levels of 15 GWh. The seasonal behaviour is still apparent, but the charging behaviour is more gradual than that of the minimum PEIS solution. In addition, the faster charging and discharging cycles on top of the seasonal behaviour suggest that in this system, weekly and diurnal storage cycles have a more important role to play, contributing to the lower operational cost of this system. Finally, only four large storage systems are installed here, with the other TES systems playing only a marginal role. Further study will have to prove whether these TES systems will be completely removed if we leave the genetic algorithm to search even more generations with larger populations, or they are actually needed and worth investing in. Whereas the previous solution was characterised by similar charging patterns, the large storage tanks' storage behaviour is clearly shifted in time. To illustrate this, *Boxbergheide* and *TermienEast* only start charging in June, whereas *ZwartbergNEast* already starts filling up in March and remains at a more or less constant level from May until end of December.

Figure 12 again shows a more detailed picture of the heat flows. The behaviour looks largely the same as in Figure 10, the difference being in the scale of the graphs, which shows that the maximum power peaks are considerably lower than those in the minimum PEIS design. This is a result of the smaller STC systems installed here.

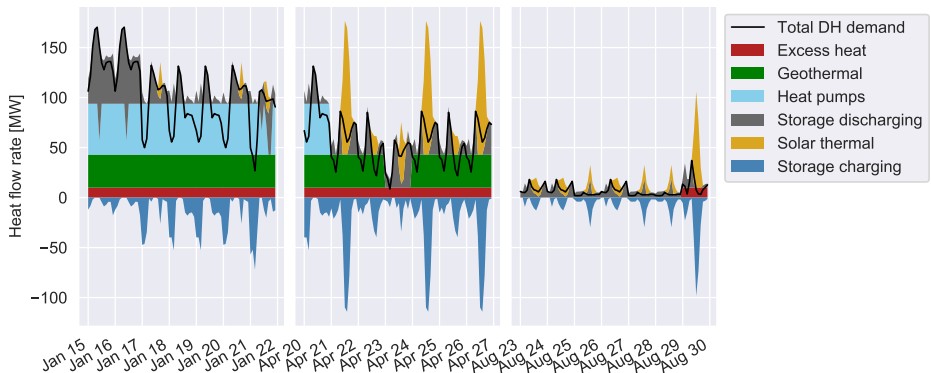

**Figure 12.** Detailed heating power plot of the intermediate PEIS design for a winter, spring and summer week.

### 3.2.3. Minimum LCOH, Maximum PEIS

The final highlighted design is that with a minimal LCOH (see Figure 13), but with the largest demand for primary energy from outside the network. Again, there are clear differences with regard to the previously discussed solutions: in this case, the excess heat injection is almost continuously at the maximum power level, whereas it was pushed out by solar power in the previous solutions. In this case, the installed STC area is very limited, which is witnessed by the absence of large positive power peaks.

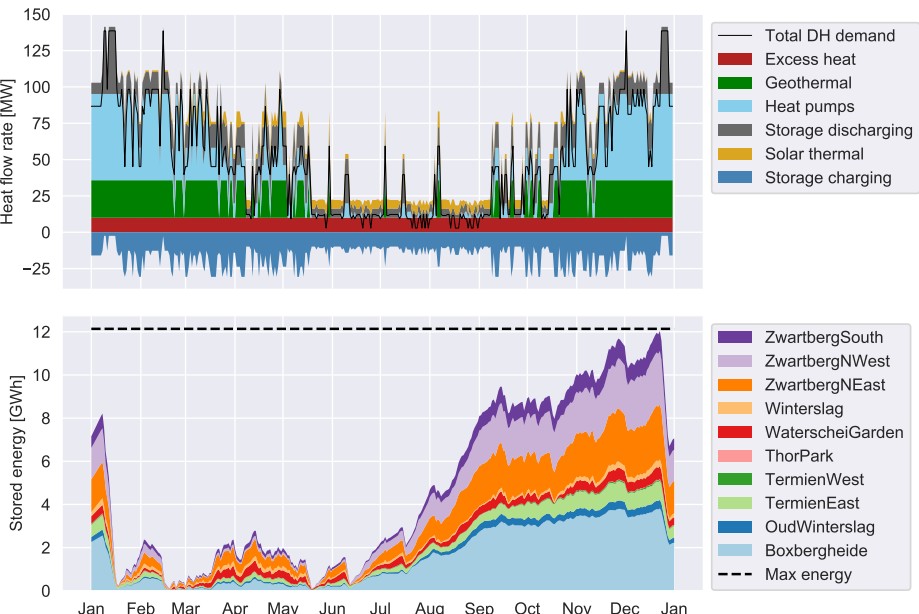

**Figure 13.** Full year time series for maximum PEIS design.

However, the TES volume is hardly smaller than that of the intermediate solution. It appears that a combination of the almost constant excess heat supply with the limited solar power injection makes it economically interesting to still have a decent amount of storage available in the network. In comparison to the intermediate solution, the average SoC is lower, which means smaller heat losses from the storage systems. Again, even though there is a clear seasonal charging pattern, there is an emphasis on shorter charging cycles to minimise operational costs.

Figure 14 shows the behaviour of the maximum PEIS system on a smaller time scale. Note the substantially lower solar thermal peaks compared to Figures 10 and 12. Another evident difference is the continuous injection of excess heat into the system, even during summer. Finally, we see that the heat pump is used more often during the spring week and even during the summer week.

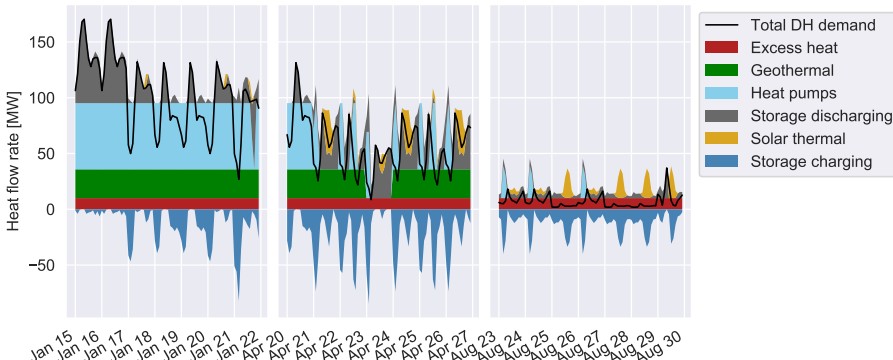

**Figure 14.** Detailed heating power plot of the maximum PEIS design for a winter, spring and summer week.

### 3.2.4. Comparison of Aggregated Results

To conclude this section on the three highlighted design solutions, Figure 15 summarises the heat delivery by the different heat sources in the network and compares these number to the total demand and storage losses. The difference between those two columns (sources vs. demand and losses) only differ by the amount of heat lost in the network. This figure makes clear that these network heat losses only make up a very limited fraction of the total heat balance. Although this result seems alarming at first, it can be explained by the fact that only the transmission network is modelled and the distribution network is omitted. Typically, these transmission pipes are much more efficient because of their wider diameter and the relatively low loss surface compared to the high mass flow rates that flow through these pipes. In addition, further analysis of the charging and discharging behaviour in these three highlighted solutions shows that an annual round-trip efficiency between 83% and 90% is achieved with the current storage model.

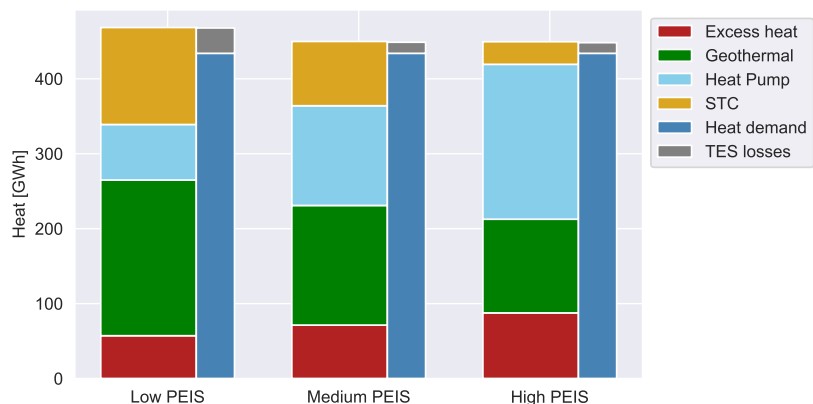

**Figure 15.** Summary of heat delivery for the three highlighted solutions.

### 3.3. Influence of Network Temperatures

This section compares the results of different temperature scenarios to the reference scenario with a 55/35 °C temperature regime. Figure 16 shows the differences between the resulting Pareto-optimal solutions with respect to the LCOH and PEIS objective functions.

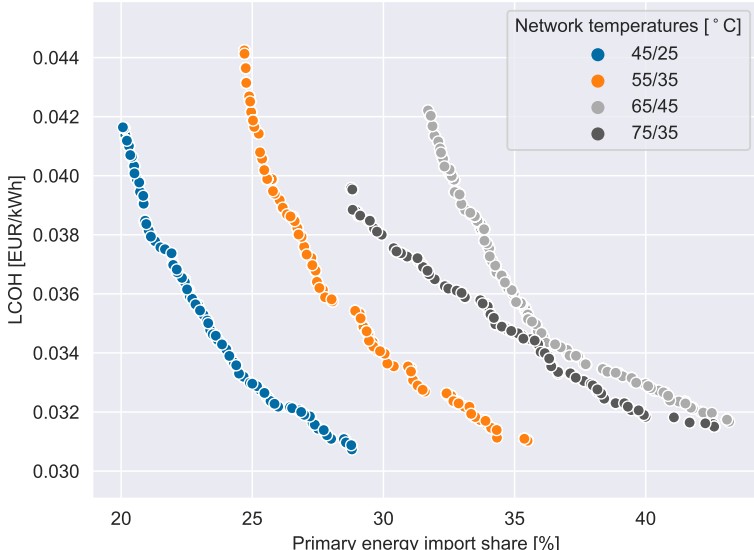

**Figure 16.** Comparison of the Pareto-optimal solutions for the different temperature scenarios. The excess heat cost is fixed at 15 EUR/MWh.

There is an obvious shift on the PEIS-axis, correlated with the average network temperature. This shift can most probably be mainly attributed to an increase in the COP (Coefficient of Performance) of the heat pumps and the geothermal plant with lower average network temperatures. The heat losses will also diminish with lower average temperatures, but given the limited influence of these losses, even more so with regard to the primary energy import into the network, it is expected that their role in the PEIS shift is of minor importance.

The 75/35 °C case is the only scenario with a 40 °C temperature difference, and this translates into a lower average LCOH. This is a result of the smaller pipes required for the same heating power transport compared to a system with a lower temperature difference, and by a more efficient utilisation of the same TES volume or STC area. The 20 °C $\Delta T$ scenarios are characterised by very similar LCOH ranges, but greatly differ in PEIS, where the reduced heat loss from the network and the increased COP of heat pump-based conversion systems are thought to be the main reason for these differences.

Considering the gas price in Belgium is currently between 0.036 and 0.039 EUR/kWh depending on the type of customer (Belgian Commission for Electricity and Gas Regulation CREG [35])—including the commodity price and the network cost, excluding taxes and levies, and disregarding the costs for a gas boiler or the loss of efficiency of a realistic heating system—we see that the current business as usual situation is already close to the LCOHs encountered in the future DES design. Compared to those numbers alone, already many of the proposed DES designs are actually cheaper than individual gas-based heating. At least, the system costs are clearly in the same order of magnitude. To make an honest comparison, the cost of the distribution networks (not considered in this study), but also the investment and maintenance of gas boilers would have to be included in the analysis. Hence, these results need to be interpreted carefully.

Figure 17 shows the Pareto fronts in terms of $CO_2$ intensity and LCOH for the different temperature scenarios. The carbon intensities for the network vary between 0.02 and 0.045 kg/kWh. For comparison, the specific $CO_2$ emissions for combustion of natural gas is 0.2 kg/kWh [36]. Even without accounting for the efficiency of a natural gas boiler, it can be seen that the $CO_2$ emissions of the studied designs are considerably lower. Clearly, Figure 17 is almost identical to Figure 16, except for the different *x*-axis. We refer to Appendix A.2.5 for a description of the time variation of the $CO_2$ intensity of the electricity consumption. Another important cost to include would be the current carbon tax. Moreover, this type of analysis could be used to determine which carbon tax levels would be needed to push the market towards more renewable technologies.

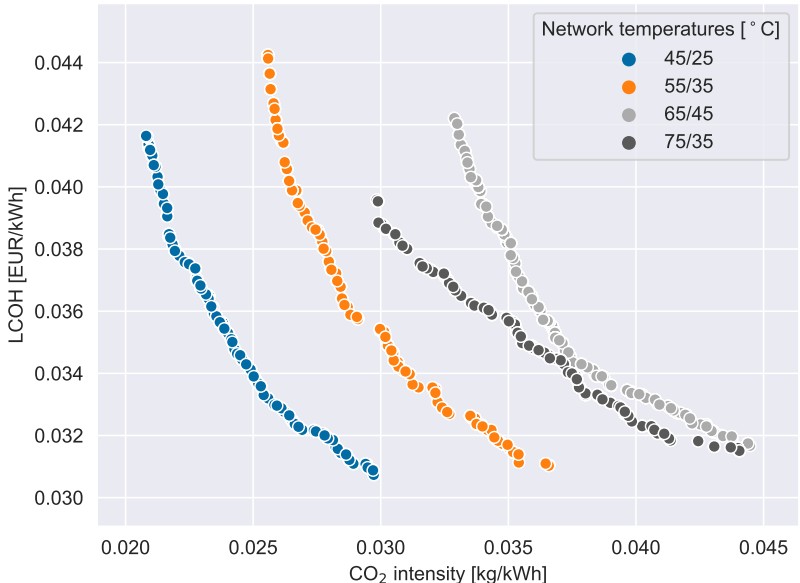

**Figure 17.** CO$_2$ intensity versus LCOH Pareto fronts for different temperature scenarios.

Figure 18 shows the relationships between the installed TES and STC sizes and their influence on LCOH and PEIS. In this figure, we see a more or less linear correlation for all temperature levels except the reference scenario. The 75/35 °C scenario has a slightly smaller TES volume distribution, but the spread on the STC area is more or less the same.

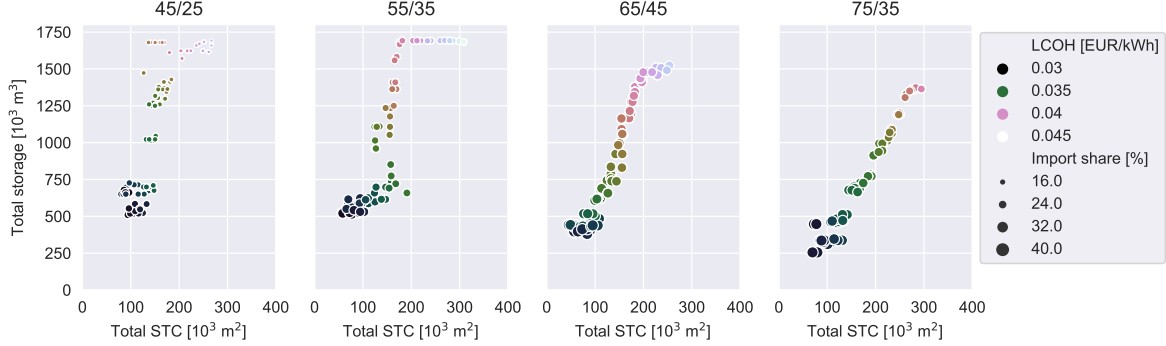

**Figure 18.** Correlation plot between total TES volume and STC area for GenkNet for different network temperatures. All plots have a fixed excess heat cost of 15 EUR/MWh.

### 3.4. Influence of Cost of Excess Heat

The differences in the Pareto fronts of scenarios with different excess heat cost—but with fixed network temperatures of 55/35 °C—are shown in Figure 19. The differences between the fronts are very subtle, and on the first sight they seem to be mostly coinciding. On closer inspection, the front with 20 EUR/MWh for excess heat is usually on top, indicating a slightly higher LCOH for the same PEIS, although in the higher LCOH range this is not always the case. In addition, the Pareto front of this highest excess heat cost stretches out further to the lower right than the other fronts. This means that lower cost solutions can be reached by importing more energy when the excess heat cost is higher, however the differences are very small.

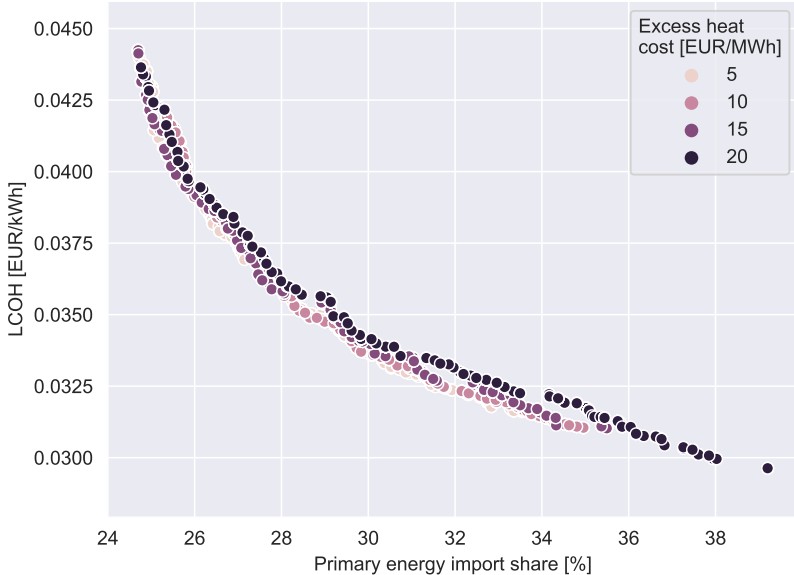

**Figure 19.** Comparison of the Pareto-optimal solutions for the different excess heat cost scenarios. The network temperatures are fixed at 55/35 °C.

These results suggest that the excess heat cost is of minor importance to the solution, at least for the cost range studied here. Indeed, the cost of excess heat is small compared to the total levelised costs encountered in the solution space. We expect to see larger differences when the excess heat cost approaches or exceeds the current system LCOHs. Moreover, the importance of the investment to connect the industrial excess heat sources is not to be forgotten. Because the availability of excess heat requires a district heating connection with a substantial investment cost, we expect that the point at which excess heat is no longer included in the optimal design will be already encountered at a lower excess heat price than the current system LCOH levels. Currently this tipping point at which excess heat is no longer chosen as a source of heat has not been encountered yet.

The main investment cost consideration influenced by the price of excess heat is the investment in the connection between the backbone and the node *GenkZuid* at which the excess heat is available. Therefore, we study the chosen diameter of this connection. Figure 20 shows the diameter for different energy ranges and excess heat costs, and it is clear that for the highest excess heat cost, the installed diameters are substantially smaller. Furthermore we see that the low energy import solutions always have a wider connection with the excess heat node than the ones with a higher energy range. Strangely, again we see an increasing trend in the diameters of the excess heat connection for the three lowest excess heat prices, which appears to be contradictory to the intuition that one would invest less to get access to a more expensive commodity, however also operational costs play.

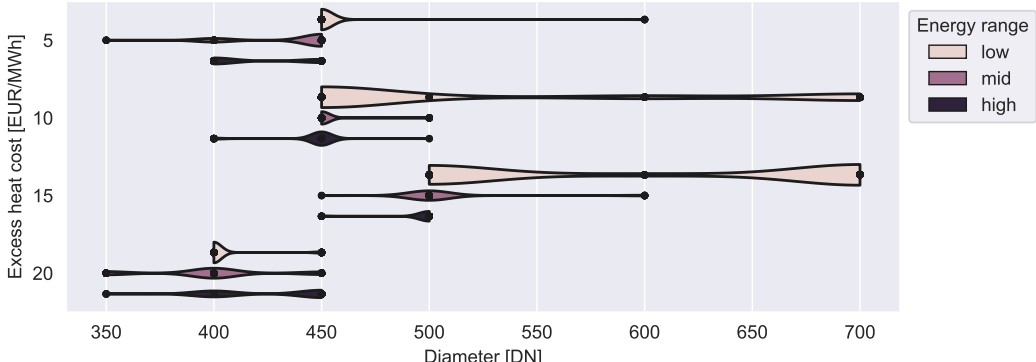

**Figure 20.** Influence of the excess heat cost on the diameter of the connection between the excess heat injection node and the rest of the network.

In summary, the cost of excess heat does not have a very clear influence on the design choices, except for the diameter of the excess heat node connection.

### 3.5. Calculation Times

For the seven calculated scenarios, the average calculation time for an entire genetic algorithm run (with 100 generations of 60 individuals) took 17.7 h. The algorithm was run on a Dell Precision 7920 Tower workstation with two Intel Xeon Silver 4116 processors (2.1 GHz–3.0 GHz Turbo, 12 cores) and 64 GB 2666 MHz DDR4 RAM. Hence, the average operational optimisation (a single evaluation) took 127.4 s, incorporating the fact that 12 processes were run simultaneously. Because the solver usually identifies an infeasible design much quicker, the average `modesto` calculation time for a feasible design is expected to be higher. However, a large spread on the calculation times was observed, where the problem was typically harder to solve as the system design got closer to its feasibility boundaries—that is, when the components were less oversized.

The fastest genetic algorithm run was performed in 12.0 h, the slowest in 24.9 h.

## 4. Discussion

The results have mostly shown that the genetic algorithm is able to select optimised designs using `modesto` as the evaluation core to calculate the objective function values and using representative days to reduce the calculation time. However, finding generalisable conclusions or rules of thumb based on these results is difficult. Even if we could find general rules based on the presented results, more data for different kinds of networks and additional scenarios would be needed in order to really generalise. Moreover, because of the heuristic nature of a genetic algorithm, we cannot prove the optimality of the results. Nonetheless, the convergence of the Pareto-optimal solutions is an indication that at least a local, and in the best case a global optimum is approached.

In addition, further study would be needed to increase the level of detail of the optimisation problem; for example, currently we don't account for the efficiency of substations that deliver the heat from the network to the end-users. Also the distribution networks in the separate neighbourhoods are not modelled at this moment; this means the investment is considerably underestimated, and also the heat demand might increase due to extra heat losses.

The omission of the building thermal flexibility from the model is expected to have a smaller influence, given the presence of seasonality in all storage SoC profiles, even for the solution with the highest share of primary energy import. Still, it would be interesting to see if the possibility of using the already available sources of energy flexibility (see Vandermeulen et al. [37,38] for a comprehensive discussion of this topic) in thermal networks would influence the design choices made.

One of the objectives of this thesis was to optimise the location of TES systems in a network. From the highlighted solutions and the distribution of the storage sizes, a possible conclusion would be

that TES systems are preferably installed as close as possible to the heat sources in the network—that is, the heat pumps and geothermal plant. Of course, this result is biased since the STC systems also inject heat. But in general, the conclusion that heat should be stored as close as possible to where it will be generated is intuitive as it minimises transport losses.

The simultaneous optimisation of the network pipe diameters adds an interesting dimension to the results. Whereas smaller diameters are mentioned as one of the advantages of future, smart thermal networks (see Lund et al. [3]), the design results with maximal renewable and residual energy shares show that the pipe diameters actually become larger to enable large peak flow due to the redistribution of stored energy in the network and due to the high power injection by the solar thermal collectors.

A final remark must be made about the convergence of the genetic algorithm. The comparison of the solutions with different excess heat cost showed some counter-intuitive results, which could be explained by a possible incomplete convergence of the genetic algorithm. Upon closer investigation of the evolution of the hypervolume indicator (not included in the results of this paper), we see that the increase of the indicator flattens, indicating (near) convergence of the algorithm. Still, sometimes a sudden jump in the indicator—usually because of an accompanying jump in the Pareto front—is observed. Hence, it is possible that the most optimal Pareto front has not been reached yet. To be sure of the convergence, the genetic algorithm should be run with an even higher number of generations. On the other hand, the trade-off must be made between the optimality of the outcome, compared to the additional computation time needed to reach it. A similar trade-off is seen in the solution of MILPs, where an optimality gap is allowed to greatly reduce the solution time, at the expense of a slightly weaker certainty about the optimality of the solution.

## 5. Conclusions and Further Recommendations

This paper describes and illustrates an integrated design and control optimisation algorithm. For the evaluation of a single design, a Python toolbox named `modesto`, which implements a full year linear optimal control problem is used, in conjunction with an optimised representative days method to reduce the temporal complexity of the evaluations. By comparing designs using an optimal control problem, they can be compared objectively with regard to their maximum potential, and the comparison becomes independent of the efficiency of chosen control strategies. This optimal control problem is used as the evaluation core of a genetic algorithm, in which the design parameters are varied. As such, a two-layer design optimisation algorithm is constructed, where the lower layer optimises the operational aspect of the proposed networks, whereas the upper layer varies the design parameters in order to find the optimum with respect to a number of predefined objective functions.

This optimisation algorithm is applied to a fictive district heating network for the city of Genk in Belgium, and the influence of the network temperatures and the excess heat cost is investigated using a number of scenarios. It is shown that the design algorithm is able to efficiently find optimal solutions with respect to multiple simultaneous objectives, and that the proposed systems are competitive with individual natural gas-based heating systems in terms of levelised cost of heating, and even outperform this gas-based non-collective reference in terms of $CO_2$ emissions. However, we were unable to derive clear rules of thumb as to where the thermal energy storage must be installed. A clear result, though, is that seasonal energy storage will be crucial for future energy systems, as ample storage volumes are selected even for the cheapest solutions with the highest primary energy import into the network.

Suggestions for further research would be to add more modelling detail, for example by modelling the heat losses in the neighbourhood distribution networks explicitly (currently, only the transport pipes are modelled without considering the additional heat losses and investment in the neighbourhood distribution grids). Modelling the entire distribution network in detail however would probably result in very complex optimisation problems. Therefore, further study of how a homogeneous distribution network with distributed injection of renewable heat (STC panels on the rooftops of every building) could be aggregated, is required. Secondly, the `modesto` framework allows for easy addition of more energy conversion and thermal energy storage models. As such,

the connection between different energy carriers (i.e., natural gas and electricity) could be strengthened with the addition of for example cobined heat and power (CHP) systems. This would also add a larger variation on the $CO_2$ objective, which is currently very strongly linked to the primary energy import into the network. As a last improvement to the optimisation toolbox, also the network temperatures could be varied, but in a deterministic way. The supply temperature could be varied as a function of the ambient temperature (heating curve), and as a pre-processing step this would maintain the linearity of the optimisation problem. However, this would require a modification of the energy storage models, which currently rely on fixed high and low temperature levels.

**Author Contributions:** Conceptualisation, B.v.d.H., A.V., R.S. and L.H.; methodology, B.v.d.H. and A.V.; software, B.v.d.H. and A.V.; verification, B.v.d.H.; writing—original draft preparation, B.v.d.H.; writing—review and editing, B.v.d.H., A.V., R.S. and L.H.; visualisation, B.v.d.H.; supervision, R.S. and L.H.

**Funding:** This research was funded by VITO grant numbers 1510475 and 1710760.

**Acknowledgments:** The authors want to acknowledge the work of Ina De Jaeger for the curation and analysis of the GIS database of Genk and for providing the building model parameters. We are grateful for the feedback on the results by Jan Diriken and Tijs Van Oevelen. Furthermore, we would like to thank Vincent Reinbold for the fruitful discussions that have contributed to this work. The work of Bram van der Heijde and Annelies Vandermeulen is funded through VITO PhD Scholarships.

**Conflicts of Interest:** The authors declare no conflict of interest.

## Nomenclature

| | |
|---|---|
| $\Delta T$ | Nominal network temperature difference [K] |
| $\dot{W}$ | Mechanical power [W] |
| $d\dot{q}$ | Heat loss rate per unit length [W/m] |
| $\dot{m}$ | Mass flow rate [kg/s] |
| $\dot{Q}$ | Heat flow rate [W] |
| $\tau$ | Economic lifetime |
| $\gamma$ | Allowed variation on the nominal network temperature difference |
| $A$ | Solar thermal collector area |
| $c_{maint,a}$ | Annual maintenance cost |
| $c_{op,a}$ | Annual operation cost |
| $c_a$ | Total annualised cost |
| $c_p$ | Specific heat of water [J/(kg K)] |
| $i$ | Interest rate |
| $I_{tot}$ | Total investment cost |
| $I_a$ | Annualised investment cost |
| $L$ | Pipe length [m] |
| $R_s$ | Symmetrical thermal pipe resistance [Km/W] |
| $T_a$ | Ambient temperature [K] |
| $T_m$ | Average solar thermal collector panel temperature |
| $T_r$ | Network return temperature [K] |
| $T_v$ | Network supply temperature [K] |
| COP | Coefficient of Performance |
| DES | District Energy System |
| DHW | Domestic Hot Water |
| GHG | Greenhouse Gas Emissions |
| LCOH | Levelised Cost of Heating |
| MILP | Mixed Integer Linear Program |
| OCP | Optimal Control Problem |
| PEF | Primary Energy Factor |
| PEIS | Primary Energy Import Share |
| PSO | Particle Swarm Optimisation |
| PTES | Pit Thermal Energy Storage |
| $R^2ES$ | Renewable and Residual Energy Sources |
| STC | Solar Thermal Collector |
| TES | Thermal Energy Storage |
| TTES | Tank Thermal Energy Storage |

## Appendix A. Operational Optimisation Models

This section provides a summary of model equations used. Because of the modular structure and the complex problems considered, giving a comprehensive overview of all optimisation variables and constraints is not possible.

### *Appendix A.1. Network Models*

The heat losses from the district heating network are calculated (see van der Heijde et al. [39]) based on a nominal supply and return temperature. These nominal temperatures are also used by all other components in the DES, such as TES and conversion systems. To maintain the model's linearity on the one hand, and to avoid infinite temperature differences at low mass flow rates (in the case of nearly mass flow-independent heat losses) on the other, the heat losses from the pipes are made a linear function of the mass flow rate. The nominal heat loss level is reached around the mass flow rate that corresponds to a pressure gradient of 80–100 Pa/m.

Pumping losses in the network are modelled with a set of linear inequality constraints. These linear segments interpolate between equidistant points on the actual pumping power curve (third-degree function of mass flow rate). Because of the inequality constraints, only branched networks can be currently modelled, and the pumping energy must be represented in the operational objective function, such that high deviations from the inequality constraints are penalised. In this case, the (nonnegative) cost for the electricity to drive the pumps is a part of the operational cost.

As such, the following model equations can be written for a pipe model:

$$\dot{Q}_{in} = L d\dot{q} + \dot{Q}_{out} \tag{A1}$$

$$\dot{m}c_p\Delta T \leq \dot{Q}_{in} \leq \dot{m}c_p(1+\gamma)\Delta T \tag{A2}$$

$$\dot{m}c_p\Delta T \leq \dot{Q}_{out} \leq \dot{m}c_p(1+\gamma)\Delta T, \tag{A3}$$

where $\gamma$ denotes the allowed variation on the temperature difference in the network to allow heat losses without violating the energy balance. We limit $\gamma$ to 5%, but the user can specify this parameter as needed. $\dot{Q}_{in}$ and $\dot{Q}_{out}$ are the respective heat flow rates at the inlet side and outlet side of the pipe system. $\Delta T$ is the design temperature difference between the supply and return side, neglecting any temperature differences between the in- and outlet side. $L$ is the length of the modelled segment. Note that $\dot{Q}$ is defined positive when heat is transported from the inlet to the outlet of the pipe, and vice versa. Heat losses $d\dot{q}$ are positive when heat is lost from the pipe to the surrounding.

The linearised heat loss model is implemented as:

$$d\dot{q}(\dot{m}) = d\dot{q}_{nom}\frac{|\dot{m}|}{0.7233\dot{m}_{max}}, \tag{A4}$$

where

$$d\dot{q}_{nom} = \frac{T_v + T_r - 2T_a}{R_s}. \tag{A5}$$

The value of $R_s$ is listed in Tables A2 and A3. 0.7233 is a factor that influences the slope of the heat loss curve, such that the nominal heat losses occur in the region around a pressure drop of 100 Pa/m.

### *Appendix A.2. Thermal Energy Conversion Components*

The energy conversion units models are based on the Energy Hub concept (see Geidl et al. [40,41] or Evins et al. [42]). The conversion of one energy form to another is modelled using a predetermined efficiency factor. While the nominal network supply and return temperature to calculate transport heat losses are fixed, a variation on the temperature difference between supply and return $\gamma$ is allowed to be able to account for heat losses in the network. This implies that also at the production side, the relation

between the design temperature difference, mass flow and heat flow rates cannot be imposed strictly, to avoid non-linear equations.

Hence, the following inequality constraints on the mass flow rate and the temperature difference form a general representation of an energy conversion system in the thermal network:

$$\dot{m}c_p\Delta T \leq \dot{Q} \leq \dot{m}c_p(1+\gamma)\Delta T, \tag{A6}$$

where $\dot{m}$ is the mass flow rate injected into the district heating system's supply and extracted from the return side by the heat generating system and $\dot{Q}$ is the thermal power injected into the network by the conversion system. This mass flow rate is heated by at least $\Delta T$, and at most $(1+\gamma) \cdot \Delta T$. These two inequalities make sure that the conversion unit is not able to inject heat into the network at zero mass flow rate, which would imply an infinite temperature difference. In addition, the maximum heat injection is limited to the nominal power of the conversion system, which is a design parameter:

$$\dot{Q} \leq \dot{Q}_{max} \tag{A7}$$

This means that even if a larger temperature difference than the nominal $\Delta T$ is used, the thermal output of the unit cannot be higher than its nominal power. There is no explicit bound on the mass flow rate, other than that following out of Equation (A6) and out of the mass flow rate limits of the pipes connected to the network node in which the conversion system is embedded. All conversion systems are assumed to be able to modulate their output 100%, unless stated differently.

Appendix A.2.1. Heat Pumps

The heat pumps considered in this study are large air source heat pumps. Its COP (Coefficient of Performance) is pre-calculated based on the variation of the ambient temperature, and based on the network temperatures:

$$\text{COP} = \frac{\dot{Q}}{\dot{W}} = \eta_C \frac{T_v}{T_v - T_a}, \tag{A8}$$

assuming a non-ideal Carnot cycle with a relative efficiency $\eta_C = 0.6$ (compared to the ideal Carnot cycle). In this equation, $\dot{W}$ is the mechanical power supplied to the heat pump in the form of electricity, and $T_v$ and $T_a$ are the respective supply and ambient temperature. The Carnot efficiency, although seemingly optimistic, was chosen in line with data from the Danish Energy Agency [43] for large heat pumps.

Appendix A.2.2. Solar Thermal Collectors

The STC model assumes a steady-state heat delivery as a function of the mass flow rate and the solar irradiance only, and was derived from norm EN 12975-2 [44]:

$$\dot{Q}_{out}(t) = A \cdot \left( \eta_0 \dot{Q}_{sol}(t) - a_1 \left( T_m - T_a(t) \right) - a_2 \left( T_m - T_a(t) \right)^2 \right) \tag{A9}$$

In this equation, $\dot{Q}_{out}$ and $\dot{Q}_{sol}$ represent the heat output and the solar irradiance on the unit surface respectively. The collector surface area is represented by $A$. $\eta_0$ is the base efficiency and $a_1$ and $a_2$ are temperature dependence parameters. Manufacturers of STC panels measure these values according to the norm mentioned before. This paper assumes Arcon Sunmark HT-SolarBoost 35/10 flat-plate collectors with an $\eta_0$ base efficiency of 0.839, an $a_1$ value of 2.46 W/(m$^2$ K) and an $a_2$ factor of 0.0197 W/(m$^2$ K$^2$) [45]. $T_a$ is the ambient temperature, whereas $T_m$ represents the mean panel temperature. We assume $T_m$ to be the average of the supply and return temperature of the network, which would correspond with a linear temperature increase along the collector pipe. When the heat output would become negative according to Equation (A9), it is set to be exactly 0 W.

Solar panels are assumed to be oriented South at a 40° tilt angle. The solar panels studied in the design optimisation are either installed in an empty field in rows (no shading effects accounted for), or mounted on the south-oriented parts of the roof of the buildings in the selected neighbourhoods. The incident solar radiation on a tilted unit area was simulated in Dymola [46] using typical meteorological year data for Brussels. The effect of the incidence angle of solar radiation on the panel on the transmission and reflection of the solar irradiance is neglected for simplicity.

Appendix A.2.3. Geothermal Heating Plant

The Danish Energy Agency [43] considers a geothermal plant for district heating in combination with an electric heat pump that assists in extracting as much heat as possible from the ground. The water is pumped up from the ground extraction well and passes through a heat exchanger to preheat the water from the return side of the district heating system. The water in the geothermal loop is then cooled by the heat pump evaporator before it is pumped back into the injection well. The heat pump condenser in its turn injects heat into the district heating loop until to the desired supply temperature is reached.

A geothermal well is designed to be operated continuously and therefore it can hardly change the output power. We predetermine the period during which the geothermal plant is (in)active, usually in the summer months, in this case the plant is shut down from the end of May until the end of September (day 150 and 270 of the year).

The coefficient of performance of the system is determined based on the temperatures of the geothermal well doublet, combined with the design temperatures of the network and a pinch temperature difference of 5 K in the heat exchanger. This leads to a non-linear system of equations, which is solved as a pre-processing step and the resulting COP is used in modesto. We will not treat the non-linear system of equations in further detail here, as they can be derived from the system lay-out in Reference [43].

Appendix A.2.4. Industrial Excess Heat

Industrial excess heat is treated as a simple heat source with a fixed cost per unit of energy. It is assumed that the industry suppliers cover the entire investment and pumping costs and that these expenses are reflected in the excess heat price. Furthermore, we assume a constant availability of the nominal heating power, without the obligation to buy all of it. Down-periods for maintenance of the industrial processes are not accounted for.

Appendix A.2.5. Electricity Considerations

The electricity used by the heat pump components is assumed to be extracted from the electricity grid. We are using Belgian electricity grid data, namely the BELPEX day-ahead prices for the year 2014. The primary energy factor (PEF) and $CO_2$ intensity for the Belgian grid have been calculated by Vandermeulen and Vandeplas [47]. We have calculated the hourly average profile as a representation and assume the same profile is valid for every day of the year. The time series of these profiles ($CO_2$ intensity, PEF and electricity price) have been summarised in Figure A1. Note that the electricity price is incorporated in the representative days selection algorithm and hence the hourly average profile is not used in the optimisation framework. The reason for this choice is that the electricity cost has a large influence on the operational objective function of the optimisation, here the real yearly variation has been chosen instead of a daily aggregated version.

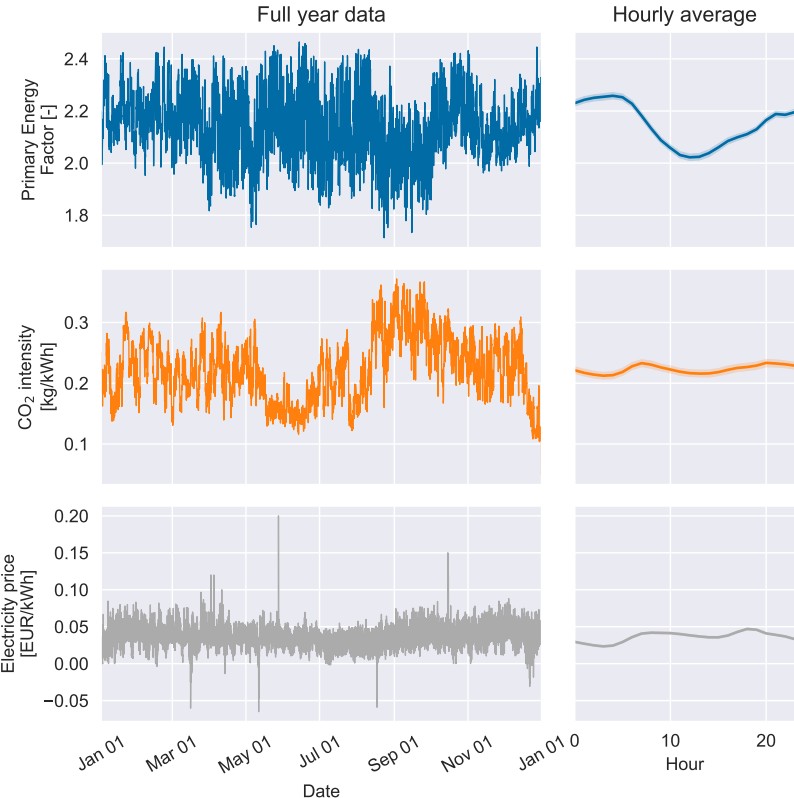

**Figure A1.** Time series and hourly average of PEF, CO$_2$ intensity and electricity cost.

The fact that these time series are not realistically correlated with the weather data is disregarded here.

*Appendix A.3. Thermal Energy Storage*

This paper considers two types of TES, namely tank and pit thermal energy storage (TTES and PTES). Both are modelled in line with the model used by Vandewalle and D'haeseleer [26], namely a perfectly stratified model (using the network supply and return temperature as the respective high and low TES temperatures), including state-dependent heat losses to the surroundings. The state of charge must be between 0% and 100%, and the (dis)charge heat flow is unconstrained.

Appendix A.3.1. Tank TES Systems

TTES systems have the advantage of better insulation over PTES systems, but due to their construction they are limited in size. In addition, the heat losses can be reduced even further by choosing an advantageous tank aspect ratio, minimising the tank surface with respect to the volume. We assume that the tank is constructed with a 0.5 m thick concrete shell, surrounded by 0.3 m of insulating material [48]. The concrete has a thermal conductivity of 1.63 W/(m K) and foam glass gravel is chosen as insulation, with a thermal conductivity of 0.095 W/(m K). Also, we assume that the wall and insulation thickness are the same for all of the tank walls. For TTES, we simply assume a cylindrical shape. Hence, the dependence of the heat losses is perfectly linear in the SoC. To minimise the surface area of the cylinder, we choose the aspect ratio $h/D = 1$.

Seasonal TTES systems are generally partly buried (or bermed) underground, which leads us to assume the average of the ambient and ground temperature as the boundary condition for the side walls of the tank. The bottom only sees the undisturbed ground temperature as a boundary condition, whereas the top of the tank is exposed to the ambient temperature.



Appendix A.3.2. Pit TES Systems

A comprehensive description about design of PTES systems can be found in Sørensen et al. [49]. Sorknæs [50] suggests a pit side and bottom wall conductivity of 0.5 W/(m·K) in case of sand, with an equivalent soil layer thickness of 2 m before the soil reaches the undisturbed ground temperature. The top insulation has a thickness of 0.24 m with a thermal conductivity of 0.104 W/(m·K). Because of the width variation at different depths, the relation between the SoC and the heat losses is no longer perfectly linear. However, it can be assumed to be linear without too large deviations. This is the result of the trapezoidal cross-section of the pit, where the base is narrower than the top edge. In this paper, a fixed shape of the storage pit is assumed, where the top width is 9 times the height, and the inclination of the sides is exactly 1 in 2 (or 26.57°), such that the bottom width becomes 5 times the height. Assuming a square top and bottom surface, the maximum considered pit volume of 200,000 m$^3$ has a height of approximately 16 m, whereas the top and bottom width are 144 m and 80 m respectively.

For the boundary temperature at the bottom of the pit, the undisturbed ground temperature $T_g$ is used. The walls of the pit are assumed to be exposed to the average of the ground temperature and the outside air temperature, given the rather limited depth, including the part of the pit above the ground. The top cover is exposed to the ambient temperature only.

## Appendix B. Techno-Economic Data

This section summarises data about costs, efficiency and lifetime of the used technologies.

*Appendix B.1. Thermal Energy Conversion Systems*

Table A1 summarises the used unit prices and investment analysis characteristics for the integrated optimal design and control algorithm. Note that cost figures per nominal power are always expressed considering the thermal power output.

**Table A1.** Characteristics of used conversion technologies.

| Name | Investment | Fixed Maint. [% inv./y] | Lifetime [y] | Ref. |
|---|---|---|---|---|
| Heat pump | 790 EUR/kW | 0.60 | 20 | [32,43] |
| Solar thermal collector | 250 EUR/m$^2$ | 0.13 | 30 | [32,51] |
| Geothermal heating | 1600 EUR/kW | 2.50 | 25 | [52] |

*Appendix B.2. Thermal Energy Storage Systems*

The unit cost of large storage systems varies with their size, hence the used cost data is represented graphically in Figure A2. The data points are derived from Schmidt and Miedaner [53]. The economic lifetime of TES systems is assumed to be 20 y, the fixed maintenance cost is 0.70 %/y with respect to the initial investment cost [32]. The economic lifetime and maintenance cost are the same for PTES and TTES.

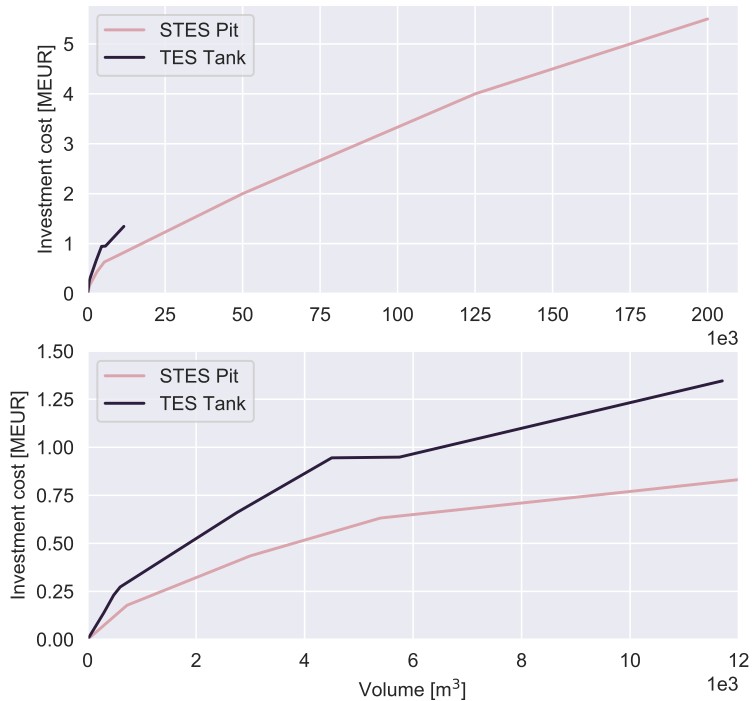

**Figure A2.** Cost evolution in function of volume for TES tanks and TES pits, using a linear interpolation between known investment costs for real systems. Data derived from Schmidt and Miedaner [53]. Note that the lower graph is representing the same data as the upper plot, but focussing on the lower volume range of the TES tank data.

*Appendix B.3. Thermal Network Pipes*

The available sizes of thermal network pipes and their dimensions are summarised in Tables A2 and A3. For the explanation of the different radii *r* and their respective accompanying diameters *d*, see Figure A3. The symmetric thermal resistance $R_s$ is calculated using the derivation by Wallentén [54] and using the conclusions of van der Heijde et al. [39].

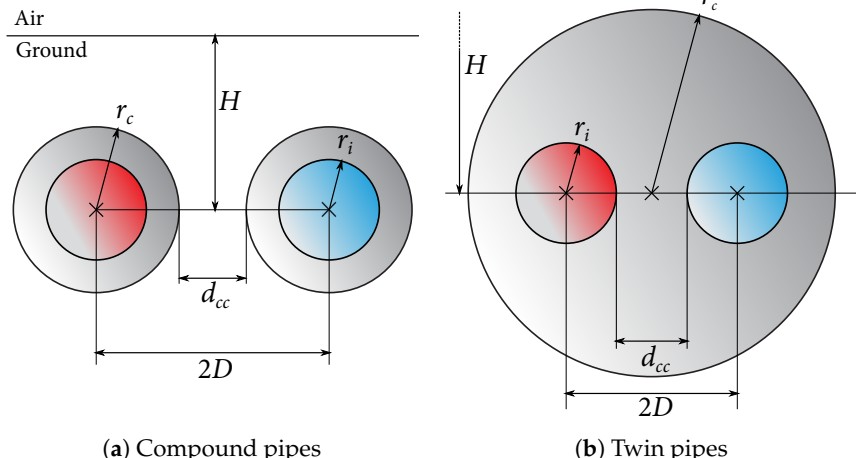

(**a**) Compound pipes　　　　　　　　　　　　　(**b**) Twin pipes

**Figure A3.** Schematic representation of the types of double pipes considered in this study with clarification of the dimensions used to calculate the thermal resistance of compound and twin pipes. The grey shaded area indicates insulation material, the red and blue shaded areas represent the water in the supply and return pipes, respectively.

**Table A2.** Dimensions of twin pipes and resulting symmetrical thermal resistance. All distances are expressed in m, the thermal resistance per unit length $R$ in $\mathrm{K\,m/W}$. All data has been retrieved from Isoplus catalogues [33].

| DN | $d_c$ | $d_i$ | $d_o$ | $d_{cc}$ | $s$ | $R_s$ |
|----|-------|-------|-------|----------|--------|--------|
| 20 | 0.125 | 0.0217 | 0.0269 | 0.019 | 0.0026 | 12.259 |
| 25 | 0.140 | 0.0273 | 0.0337 | 0.019 | 0.0032 | 11.321 |
| 32 | 0.160 | 0.0360 | 0.0424 | 0.019 | 0.0032 | 10.208 |
| 40 | 0.160 | 0.0419 | 0.0483 | 0.019 | 0.0032 | 8.591 |
| 50 | 0.200 | 0.0539 | 0.0603 | 0.020 | 0.0032 | 8.623 |
| 65 | 0.225 | 0.0697 | 0.0761 | 0.020 | 0.0032 | 7.195 |
| 80 | 0.250 | 0.0825 | 0.0889 | 0.025 | 0.0032 | 6.277 |
| 100 | 0.315 | 0.1071 | 0.1143 | 0.025 | 0.0036 | 6.199 |
| 125 | 0.400 | 0.1325 | 0.1397 | 0.030 | 0.0036 | 6.629 |
| 150 | 0.450 | 0.1603 | 0.1683 | 0.040 | 0.0040 | 5.470 |
| 200 | 0.560 | 0.2101 | 0.2191 | 0.045 | 0.0045 | 4.878 |

$d_o$ represents the outer diameter of the pipe wall, inside the insulation. $d_{cc}$ is the distance between the walls of the two pipes, such that $D = \frac{d_{cc}+d_o}{2}$. $s$ is the pipe wall thickness.

**Table A3.** Dimensions of compound single pipes and resulting symmetrical thermal resistance. All distances are expressed in m, the thermal resistance per unit length $R$ in $\mathrm{K\,m/W}$. All data has been retrieved from Isoplus catalogues [33].

| DN | $d_c$ | $d_i$ | $d_o$ | $d_{cc}$ | $s$ | $R_s$ |
|----|-------|-------|-------|----------|--------|--------|
| 250 | 0.40 | 0.2630 | 0.2730 | 0.4 | 0.0050 | 2.773 |
| 300 | 0.45 | 0.3127 | 0.3239 | 0.4 | 0.0056 | 2.437 |
| 350 | 0.50 | 0.3444 | 0.3556 | 0.4 | 0.0056 | 2.480 |
| 400 | 0.56 | 0.3938 | 0.4064 | 0.5 | 0.0063 | 2.340 |
| 450 | 0.63 | 0.4446 | 0.4572 | 0.5 | 0.0063 | 2.311 |
| 500 | 0.67 | 0.4954 | 0.5080 | 0.6 | 0.0063 | 2.024 |
| 600 | 0.80 | 0.5958 | 0.6100 | 0.7 | 0.0071 | 1.964 |
| 700 | 0.90 | 0.6950 | 0.7110 | 0.7 | 0.0080 | 1.744 |
| 800 | 1.00 | 0.7954 | 0.8130 | 0.8 | 0.0088 | 1.559 |
| 900 | 1.10 | 0.8940 | 0.9140 | 0.8 | 0.0100 | 1.426 |
| 1000 | 1.20 | 0.9940 | 1.0160 | 0.9 | 0.0110 | 1.306 |

For compound pipes, $d_{cc}$ is the distance between the outer edge of the insulation jackets of the separate pipes, or $D = \frac{d_c+d_{cc}}{2}$.

In the study of Ahlgren et al. [55], a distinction is made between inner and outer city areas. Netterberg et al. [56] seem to have consulted a similar data source. They found a linear investment cost regression:

$$I_{inner} = 2.18 \cdot d_{DN} + 308.46 \ [\mathrm{EUR/m}] \tag{A10}$$

$$I_{outer} = 1.8596 \cdot d_{DN} + 230.5 \ [\mathrm{EUR/m}], \tag{A11}$$

where $I_{inner}$ and $I_{outer}$ denote the respective investment cost per unit length for inner and outer city areas and $d_{DN}$ is the nominal diameter of the pipe in mm.

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
