# Peer review of "Integrated Optimal Design and Control of Fourth Generation District Heating Networks with Thermal Energy Storage"

_energies, doi:10.3390/en12142766_

Round 1
Reviewer 1 Report
The topic and aim of the research is clearly defined and determined.
The Introduction is too long and unneeded to be divided into two sections (1.1. Previous studies on district energy system desig, 1.2. Novelty and contribution). It presents unnecessary information and need to be shortened.
Fig. 1 - color scale illegible. Please improve the quality of the flowchart in Fig. 3.
The Authors clearly indicated the contribution and limitations of their work.
Author Response
Please see the attachment for response to all three Reviewers' comments.

Reviewer 2 Report
The authors appropriately introduce the topic by understanding the objective of the work. The presentation of the various authors who carried out district energy system (DES) studies through mixed integer linear program (MILP) within thermal and electrical grid (TES) is clear, but could be investigated with other studies by other researchers who have previously studied optimization and design of district energy systems (DES).
Energy users could be described better to highlight their energy characteristics, namely electrical, thermal and cooling consumption.
Regarding to the definition of the model and to understand better the work carried out, at least in the crucial points, it would have been appropriate to write the most important parts in a more extended way such as: decision variables, constraints of the components, constraints of the district heating and cooling network , constraints of the thermal storage and constraints of the energy balances.
It would have been necessary to introduce a superstructure scheme of all the various components of the district energy generation system.
The three objective functions, (annualizedtotal costs, primary energy import and CO2 emissions) could be more widely exposed.
Finally I would like to point out to the authors that the introduction of some results table would make the article more understandable.
Author Response
Please see the attachment for the responses to all three Reviewers' comments.

Reviewer 3 Report
The contribution „integrated optimal design and control of fourth generation district heating networks with thermal energy storage” is interesting, innovative and relevant. It is also well written and well structured, but significant compaction is required.
On the other hand some relevant information/discussion is missing or some assumptions might be chosen with levity.
- What is the overall energy demand of the DH system
- What are the thermal losses of the pipes and of the TES for the different scenarios. Are these realistic.
- How is the TES modelled, ideally stratified, thermal losses? Is this justified. Are the losses realistic? The cost curves for the TES seem to be in an expectable range, but the trend for the tank with the plateau seems not to be justified
- The heat pump model is a very simple one, and the COP seems to be very optimistic.
- Service life of components. >30 yrs? it makes a difference to have 30, 40 or 50 yrs. A heat pump might be replaced after 15 to 20 yrs while the tank TES should have a durability of 50 yrs.
It might be good to compare some cases to more detailed results in order to increase confidence on the results. However, if the method and not the results (in absolute values) are in the focus of the paper, that might be ok. However, this should be clearly stated.
It would be good to motivate the decision for the range of TES volume and SC area.
In order to gain some space, less relevant parts should be omitted and consider to delete all not highly relevant figures such as fig. 1, fig 3 (or at least try to present it more compact), fig 8 (maybe better a table), fig. 11. Do we need all figures in section 3.2.2 and 3.2.3?
The entire chapter about the pipes seems to be of lower relevance and might be shortened, too.
Readability of Figure 12 (14 and 16, if not omitted) would be better with daily average data instead of hourly.
In the discussion, line 555. The position of the TES could be also beneficial close to where the energy is consumed (not only where it is produced). But this is anyhow a quite trivial conclusion and to derive it, less sophisticated approaches might be sufficient.
Delete “it is fair to say”.
In the introduction some relevant literature about TES might be mentioned as in Dahash, Ochs, Bianchi Janetti and Streicher, Advances in seasonal thermal energy storage for solar district heating applications: A critical review on large-scale hot-water tank and pit thermal energy storage systems, Applied Energy, vol. 239, pp. 296-315, 2019.
Author Response
Please see the attachment for the response to all three Reviewers' comments.

Round 2
Reviewer 3 Report
my comments were considered in an appropriate way. Thank you.